# Bayesian-guided Label Mapping for Visual Reprogramming

**Chengyi Cai**[1] **Zesheng Ye**[1] **Lei Feng**[2] **Jianzhong Qi**[1] **Feng Liu**[1]*

[1]The University of Melbourne    [2]Singapore University of Technology and Design

{chengyi.cai1,zesheng.ye,jianzhong.qi}@unimelb.edu.au

feng_lei@sutd.edu.sg    fengliu.ml@gmail.com

## Abstract

*Visual reprogramming* (VR) leverages the intrinsic capabilities of pretrained vision models by adapting their input or output interfaces to solve downstream tasks whose labels (i.e., downstream labels) might be totally different from the labels associated with the pretrained models (i.e., pretrained labels). When adapting the output interface, label mapping methods transform the pretrained labels to downstream labels by establishing a gradient-free one-to-one correspondence between the two sets of labels. However, in this paper, we reveal that one-to-one mappings may overlook the complex relationship between pretrained and downstream labels. Motivated by this observation, we propose a ***Bayesian-guided Label Mapping*** (BLM) method. BLM constructs an iteratively-updated probabilistic label mapping matrix, with each element quantifying a pairwise relationship between pretrained and downstream labels. The assignment of values to the constructed matrix is guided by Bayesian conditional probability, considering the joint distribution of the downstream labels and the labels predicted by the pretrained model on downstream samples. Experiments conducted on both pretrained vision models (e.g., ResNeXt) and vision-language models (e.g., CLIP) demonstrate the superior performance of BLM over existing label mapping methods. The success of BLM also offers a probabilistic lens through which to understand and analyze the effectiveness of VR. Our code is available at https://github.com/tmlr-group/BayesianLM.

## 1 Introduction

Repurposing pretrained models from data-rich domains [6, 28, 59] has emerged as an effective strategy to address downstream tasks without re-training a task-specific model. For visual tasks, *visual reprogramming* (VR) [3, 4, 49, 52]–also called adversarial reprogramming [12, 38, 48]–repurposes a pretrained model for downstream tasks without changing the model. In particular, VR (full task setup detailed in Appendix A) modifies the model's input interface by adding trainable noise patterns to the images of downstream tasks. Since pretrained and downstream tasks typically have distinct label spaces, a *label mapping* (LM) function is needed to map outputs of the pretrained models to downstream labels. Often, existing VR methods adopt a gradient-free one-to-one LM [4, 12, 48], avoiding the computational cost of training fully-connected output layers through backpropagation.

However, we find that a one-to-one LM overlooks the complex many-to-many relationship between pretrained and downstream labels, which may limit the performance of VR. In Figure 1, we repurpose a model pretrained on ImageNet [46] for downstream classification tasks using a one-to-one LM strategy [4], and present statistical results. The two subfigures Figure 1a and Figure 1b, illustrate drawbacks from the perspectives of individual images and the entire dataset, respectively. Figure 1a shows the distribution of logits (i.e., model output before the softmax layer) for the most likely

---

*Correspondence to Feng Liu (fengliu.ml@gmail.com)

38th Conference on Neural Information Processing Systems (NeurIPS 2024).

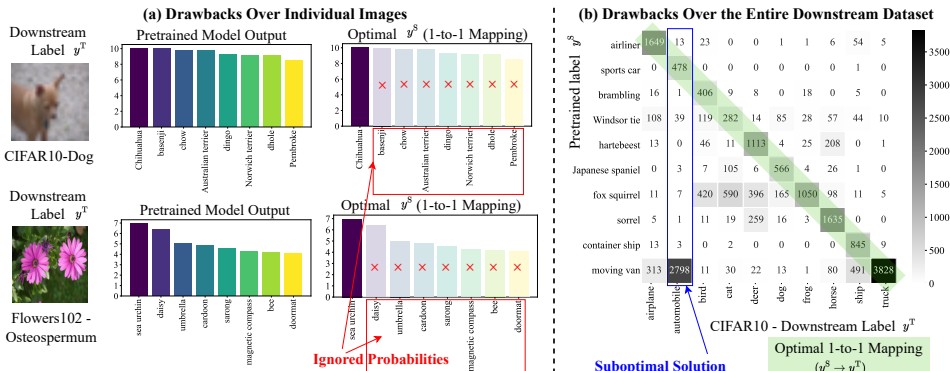

Figure 1: Drawbacks of one-to-one LM from the perspectives of (a) individual images and (b) the entire dataset. An ImageNet-pretrained classifier is reused in downstream tasks. In (a), images 'Dog' and 'Osteospermum' from downstream tasks are mapped into only one pretrained label, respectively, ignoring other probabilities. In (b), the distribution of [predicted pretrained label $y^S$, ground-truth downstream label $y^T$] pairs reveals the existence of suboptimal solutions, where 'Automobile' cannot be paired with the optimal pretrained label 'Moving Van', which has already been mapped to 'Truck'.

predicted pretrained labels of two images from downstream tasks: a 'Dog' image from CIFAR10 [30] and an 'Osteospermum' image from Flowers102 [41]. For the 'Dog' image, multiple pretrained labels like 'Chihuahua', 'Basenji'–*subclasses* of dogs–receive high logits. Similarly, for the 'Osteospermum' image, pretrained labels such as 'Sea Urchin', 'Daisy', which *share similar features*, also score high. Despite these connections, the one-to-one LM retains only the label with the highest logit, suggesting the *probabilities of other related labels are ignored*. Figure 1b shows the frequency distribution of the predicted pretrained labels and the ground-truth downstream labels of downstream samples, with the diagonal representing the results derived from one-to-one LM. The 'Automobile' class from CIFAR10, for example, can no longer be paired with the optimal pretrained label 'Moving Van', which has already been greedily mapped to the label 'Truck', implying *suboptimal label assignments*.

The above observation motivates us to go beyond these binary mappings. In Section 3, we replace the one-to-one LM function with a probabilistic LM matrix. Each matrix element is a real number that quantifies the relationship between a pretrained label and a downstream label, updated iteratively during VR optimization. This allows predictions for each downstream sample to consider diverse contributions from all pretrained labels, enabling a flexible many-to-many mapping strategy.

Specifically, we present *Bayesian-guided label mapping* (BLM) in Section 4, which assigns values to elements in the probabilistic LM matrix based on Bayesian conditional probabilities, derived from the joint distribution of the predicted pretrained labels on downstream tasks and the ground-truth downstream labels. We further extend BLM to BLM+, which aggregates *top-$K$ predicted probabilities* instead of using a single predicted label when estimating the joint distribution, accounting for uncertainty in the predictions. We also provide a theoretical analysis that justifies the potential of probabilistic many-to-many LM to outperform deterministic one-to-one LM.

To show the effectiveness of BLM, experiments are conducted on 12 widely used datasets, with BLM and BLM+ being applied to different input VR methods–padding and watermarking–on pretrained ResNet and ResNeXt (see Section 5). The ablation study and parameter analysis are also included, along with visualization results and discussions of why VR is effective. BLM and BLM+ are also applied to vision-language models (see Appendix L) to demonstrate their general applicability.

In summary, both theoretical analysis and empirical findings (Tables 1-2) provide compelling evidence that BLM and BLM+, grounded in Bayesian principles, facilitate VR to leverage pretrained knowledge for diverse downstream tasks. Beyond performance improvement, BLM and BLM+ offer insights into understanding the effectiveness of VR (Figures 3-4): revealing the relations between pretrained and downstream label spaces may guide future studies into more interpretable VR methods.

## 2 Related Works

**Model Reprogramming.** Among cutting-edge transfer learning methods (see Appendix B), model reprogramming introduces an efficient learning framework for adapting models pretrained on large-

scale data to downstream tasks constrained by limited resources [7]. By changing the input or output interfaces (i.e., input or output space) purposefully, while preserving the integrity of the pretrained model, knowledge can be reused on new tasks, sidestepping exhaustive finetuning of the model.

Many recent studies focus on repurposing diverse pretrained models for downstream tasks, including pretrained vision models [1, 4, 38, 48, 49, 52] such as ResNet [17] and ViT [11], language models [15, 50] such as BERT [24], acoustic [21, 60, 61] and graph models [23]. Such repurposing encompasses several types: cross-modal (e.g., from voice to time-series [61], or vision to text [38]), different tasks within the same modality (e.g., from image classification to out-of-distribution detection [52]), and different domains within the same task (e.g., from ImageNet to medical images [48]).

**Prompting and Input VR.** Prompting incorporates meticulously designed prompts (additional parameters) into pretrained models with specific architectures to utilize pretrained models in downstream tasks. Leveraging ViT, VPT [22] integrates prompts alongside image embeddings, while EEVPT [16] further enhances VPT by embedding parameters within self-attention layers. TransHP [53] additionally learns prompt tokens to encode coarse image categories. In vision-language models such as CLIP [45], besides text-prompting methods such as CoOP [68] and CoCoOP [67], models like MaPLe [25] also learn layer-specific mapping functions that bridge vision and text.

Slightly different from prompt tuning, input VR offers a model-agnostic approach by introducing trainable noise to images in the input space before feeding those images into pretrained models. This process does not impact the visual effect of the images. Two prevalent techniques are padding-based VR and watermarking-based VR. Padding-based models [4, 12, 48, 49] preserve the integrity of images while introducing trainable noise patterns to the outer frames around images, whereas watermarking-based models [1, 3, 42, 52] train noise patterns that overlay the images.

**Output Mapping for VR.** Because pretrained labels and downstream labels are often different, relying solely on input VR may be insufficient for downstream tasks. To bridge this gap, output mapping methods are introduced to facilitate alignment between different label spaces. Mainstream approaches include deep learning-based and statistical inference-based (i.e., gradient-free) LM methods. Deep learning-based methods insert a learnable fully connected layer to connect pretrained and downstream labels [27, 49]. However, for tasks with large label spaces, the additional model layers would result in extra training costs, potentially canceling the efficiency advantages of VR.

As for gradient-free LM methods, *random label mapping* (RLM) [12] establishes mappings between an equal number of randomly selected pretrained labels and downstream labels, masking out other unused ones. *Frequent label mapping* (FLM) [48] selects optimal one-to-one mappings using a greedy approach based on the number of pairs between pretrained and downstream labels. *Iterative label mapping* (ILM) [4] extends FLM by updating mappings at each epoch, refining the output label mapping as input VR patterns evolve. As depicted in Figure 1, these one-to-one mappings *overlook potential probabilities* and lead to *suboptimal solutions*. We propose BLM to address these issues.

## 3   Problem Formulation

**Problem Setup**. Consider a pretrained task with input and output variables $X^\mathrm{S}$ and $Y^\mathrm{S}$, jointly defined over $\mathcal{X}^\mathrm{S} \times \mathcal{Y}^\mathrm{S}$, where $\mathcal{X}^\mathrm{S} \subseteq \mathbb{R}^{d_\mathrm{S}}$ has the input dimensionality $d_\mathrm{S}$ and $\mathcal{Y}^\mathrm{S} = \{1, \ldots, k_\mathrm{S}\}$. We have a pretrained classifier $f_\mathrm{pre} : \mathcal{X}^\mathrm{S} \mapsto \mathbb{R}^{k_\mathrm{S}}$ producing a logits vector $f_\mathrm{pre}(x^\mathrm{S}) \in \mathbb{R}^{k_\mathrm{S}}$ for each $x^\mathrm{S} \in \mathcal{X}^\mathrm{S}$. For a downstream task with input and output variables $X^\mathrm{T}$ and $Y^\mathrm{T}$ defined over $\mathcal{X}^\mathrm{T} \times \mathcal{Y}^\mathrm{T}$, where $\mathcal{X}^\mathrm{T} \subseteq \mathbb{R}^{d_\mathrm{T}}$ has the input dimensionality $d_\mathrm{T}$ and $\mathcal{Y}^\mathrm{T} = \{1, \ldots, k_\mathrm{T}\}$, VR seeks to adapt $f_\mathrm{pre}$ to the downstream task without modifying its parameters. To achieve this, VR introduces two functions: 1) input VR function $f_\mathrm{in}(\cdot|\theta) : \mathcal{X}^\mathrm{T} \mapsto \mathcal{X}^\mathrm{S}$ with learnable parameters $\theta$ that converts downstream inputs for compatibility with $f_\mathrm{pre}$; and 2) output LM function $f_\mathrm{out}^\omega(\cdot) : \mathbb{R}^{k_\mathrm{S}} \mapsto \mathbb{R}^{k_\mathrm{T}}$ that aligns the output logits of $f_\mathrm{pre}$ with the downstream label space by a transformation $\omega$. Concretely, given a training dataset $\mathcal{D}^\mathrm{T} = \{(x_i^\mathrm{T}, y_i^\mathrm{T})\}_{i=1}^n$ with $n$ training samples drawn from $\mathcal{X}^\mathrm{T} \times \mathcal{Y}^\mathrm{T}$ for the downstream task, the training objective of VR can be formulated as:

$$\min_{\theta \in \Theta} \frac{1}{n} \sum_{i=1}^n \ell(y_i^\mathrm{T}, (f_\mathrm{out}^\omega \circ f_\mathrm{pre} \circ f_\mathrm{in})(x_i^\mathrm{T}; \theta)), \tag{1}$$

where $\ell$ is a loss function, and $f_\mathrm{out}^\omega \circ f_\mathrm{pre} \circ f_\mathrm{in}$ denotes the composition of input VR, pretrained model and output LM. In this study, we focus on *gradient-free* LM, where $f_\mathrm{out}^\omega$ does not introduce additional trainable parameters but strategically leverages $f_\mathrm{in}$ and $f_\mathrm{pre}$ to determine $\omega$.

**Modeling Existing LM**. As mentioned, $f_{\text{out}}^{\omega}$ serves to find a mapping between each $y^{\text{S}} \in \mathcal{Y}^{\text{S}}$ and $y^{\text{T}} \in \mathcal{Y}^{\text{T}}$. This can be achieved by constructing an output label transformation $\omega$ such that for each downstream sample $x_i^{\text{T}}$, its label $\hat{y}_i^{\text{T}}$ is predicted by $\arg \max \text{softmax}(\tilde{y}_i^{\text{T}})$, with:

$$\tilde{y}_i^{\text{T}} \equiv \begin{bmatrix} \tilde{y}_i^1 \\ \vdots \\ \tilde{y}_i^{k_{\text{T}}} \end{bmatrix} = f(x_i^{\text{T}})^{\top} \cdot \omega = \begin{bmatrix} f(x_i^{\text{T}})_1 & \cdots & f(x_i^{\text{T}})_{k_{\text{S}}} \end{bmatrix} \begin{bmatrix} \omega_{1,1} & \cdots & \omega_{1,k_{\text{T}}} \\ \vdots & \ddots & \vdots \\ \omega_{k_{\text{S}},1} & \cdots & \omega_{k_{\text{S}},k_{\text{T}}} \end{bmatrix}, \qquad (2)$$

where $f(x_i^{\text{T}})$ is shorthand for $(f_{\text{pre}} \circ f_{\text{in}})(x_i^{\text{T}}; \theta)$. $\omega$ can be updated iteratively [4] with input VR. A deterministic one-to-one relation between $\mathcal{Y}^{\text{S}}$ and $\mathcal{Y}^{\text{T}}$ implies only a *single* "correct" $y^{\text{S}} \in \mathcal{Y}^{\text{S}}$ exists for each $y^{\text{T}} \in \mathcal{Y}^{\text{T}}$. Formally, $\omega$ in Eq. (2) is a binary matrix, where just a *single* element $\omega_{j,k}$ is set to 1 in each column of $\omega$ (i.e., $\omega \in \{0,1\}^{k_{\text{S}} \times k_{\text{T}}}$ satisfying $\sum_{j=1}^{k_{\text{S}}} \omega_{j,\cdot} = 1$).

**Our Probabilistic LM.** Considering aforementioned drawbacks of one-to-one mappings, we propose a probabilistic LM for VR, assigning real values to all elements in $\omega$ (i.e., $\omega \in [0,1]^{k_{\text{S}} \times k_{\text{T}}}$ satisfying $\sum_{j=1}^{k_{\text{S}}} \omega_{j,\cdot} = 1$). Each element $\omega_{y^{\text{S}}, y^{\text{T}}}$ quantifies the relationship between $y^{\text{S}} \in \mathcal{Y}^{\text{S}}$ and $y^{\text{T}} \in \mathcal{Y}^{\text{T}}$. This acknowledges contributions from all pretrained labels for the prediction of downstream samples. The flexible many-to-many LM implies the inherent complexity in label correspondence. In Section 4, we investigate how to assign values to our probabilistic LM based on Bayes' theorem.

# 4 Bayesian-guided Probabilistic Label Mapping (BLM)

## 4.1 Method Demonstration

**Interpreting** $p(Y^{\text{T}}|X^{\text{T}})$. The objective of VR is to maximize $p(Y^{\text{T}}|X^{\text{T}})$ defined over the downstream task space. By using the law of total probability, we can express $p(Y^{\text{T}}|X^{\text{T}})$ as

$$p(Y^{\text{T}}|X^{\text{T}}) = \sum_{y^{\text{S}} \in \mathcal{Y}^{\text{S}}} p(Y^{\text{S}} = y^{\text{S}}|X^{\text{T}}) \, p(Y^{\text{T}}|Y^{\text{S}} = y^{\text{S}}, X^{\text{T}}). \qquad (3)$$

Mirroring the structure of Eq. (2), Eq. (3) enables us to estimate $p(Y^{\text{T}}|X^{\text{T}})$ using training data $\mathcal{D}^{\text{T}}$,[2]

$$\hat{p}(Y^{\text{T}}|X^{\text{T}}) = \frac{1}{n} \sum_{i=1}^{n} \left( \sum_{y^{\text{S}} \in \mathcal{Y}^{\text{S}}} \underbrace{p(Y^{\text{S}} = y^{\text{S}}|X^{\text{T}} = x_i^{\text{T}})}_{\text{① input VR:} (f_{\text{pre}} \circ f_{\text{in}})(x_i^{\text{T}}; \theta)} \underbrace{p(Y^{\text{T}} = y_i^{\text{T}}|Y^{\text{S}} = y^{\text{S}}, X^{\text{T}} = x_i^{\text{T}})}_{\text{② output LM:} f_{\text{out}}^{\omega, y^{\text{S}}}} \right), \qquad (4)$$

where ① denotes the predicted probability of pretrained label $y^{\text{S}}$ for input $x_i^{\text{T}}$, obtained from $f_{\text{pre}} \circ f_{\text{in}}$. Essentially, ① can be viewed as the standard input VR and is orthogonal to the LM methods employed; ② represents the probability that the true downstream label $y_i^{\text{T}}$ is mapped from the predicted $y^{\text{S}}$ and input $x_i^{\text{T}}$, which amounts to estimating the output label transformation $\omega \in [0,1]^{k_{\text{S}} \times k_{\text{T}}}$. Since ① is independent of output LM, the focus now shifts to estimating ②.

**Estimating** $\omega_{y^{\text{S}}, y^{\text{T}}}$ **Using Conditional Probability**. Since $\omega_{y^{\text{S}}, y^{\text{T}}}$ is used to quantify the contributions from pretrained label $y^{\text{S}}$ to downstream label $y^{\text{T}}$, we can associate it with the conditional probability:

$$p(Y^{\text{T}} = y^{\text{T}}|Y^{\text{S}} = y^{\text{S}}, X^{\text{T}}) = \frac{p(Y^{\text{T}} = y^{\text{T}}, Y^{\text{S}} = y^{\text{S}}|X^{\text{T}})}{p(Y^{\text{S}} = y^{\text{S}}|X^{\text{T}})}. \qquad (5)$$

By applying $f_{\text{pre}} \circ f_{\text{in}}$ to $\mathcal{D}^{\text{T}}$, we can empirically estimate the *joint distribution* of $p(Y^{\text{T}} = y^{\text{T}}, Y^{\text{S}} = y^{\text{S}}|X^{\text{T}})$, then obtain $p(Y^{\text{S}} = y^{\text{S}}|X^{\text{T}}) = \sum_{y^{\text{T}} \in \mathcal{Y}^{\text{T}}} p(Y^{\text{T}} = y^{\text{T}}, Y^{\text{S}} = y^{\text{S}}|X^{\text{T}})$, and substitute them into Eq. (5). Two strategies, BLM and BLM+, are presented for these estimations in this paper. To help understanding, we include a simple example to illustrate the estimation of $p(Y^{\text{T}} = y^{\text{T}}, Y^{\text{S}} = y^{\text{S}}|X^{\text{T}})$ and $p(Y^{\text{S}} = y^{\text{S}}|X^{\text{T}})$ in Appendix C.

**BLM**. Let $f(x_i^{\text{T}}) \equiv (f_{\text{pre}} \circ f_{\text{in}})(x_i^{\text{T}}; \theta)$ denote the predicted logits obtained from the pretrained model for a given input $x_i^{\text{T}}$. We define $\hat{y}_i^{\text{S}} = \arg \max_{y' \in \mathcal{Y}^{\text{S}}} f(x_i^{\text{T}})_{y'}$ to be the predicted pretrained label for $x_i^{\text{T}}$ and $\mathbb{1}\{\cdot\}$ to be the indicator function. Starting with the joint distribution $p(Y^{\text{T}} = y^{\text{T}}, Y^{\text{S}} = y^{\text{S}}|X^{\text{T}})$, we could intuitively count the frequency of $(\hat{y}_i^{\text{S}} = y^{\text{S}} \wedge y_i^{\text{T}} = y^{\text{T}})$ to estimate:

$$\hat{p}_{\text{BLM}}(Y^{\text{T}} = y^{\text{T}}, Y^{\text{S}} = y^{\text{S}}|X^{\text{T}}) = \frac{\sum_{i=1}^{n} \mathbb{1}\{y_i^{\text{T}} = y^{\text{T}}\} \cdot \mathbb{1}\{\hat{y}_i^{\text{S}} = y^{\text{S}}\}}{n}. \qquad (6)$$

---

[2]This estimation is similar to [40], see Appendix B for more discussion.

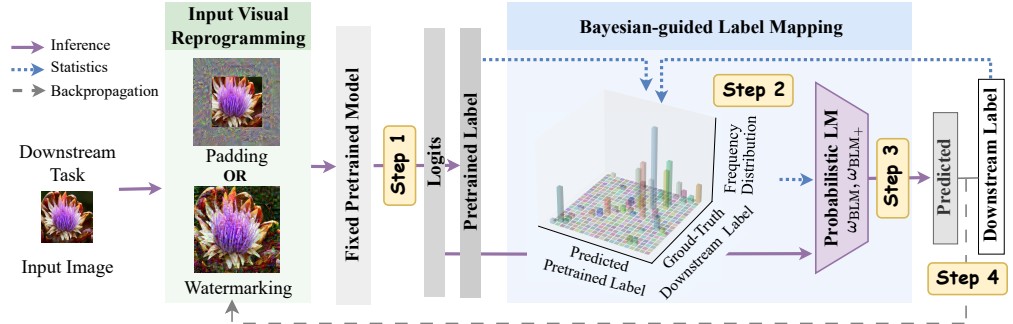

Figure 2: Learning strategy of BLM and BLM+. First, input images, incorporated with VR watermarking or padding patterns, are fed into a fixed pretrained model to obtain logits and predicted labels. Then, the true labels (of $y^{\mathrm{T}}$) and predicted labels (of $y^{\mathrm{S}}$) are used to estimate $\omega_{\mathrm{BLM}}$ or $\omega_{\mathrm{BLM}_+}$. Next, using $\omega_{\mathrm{BLM}}$ or $\omega_{\mathrm{BLM}_+}$ that reweights output logits of pretrained models for the downstream labels, the predicted results can be derived. Finally, backpropagation is performed to update the input VR.

For $p(Y^{\mathrm{S}} = y^{\mathrm{S}}|X^{\mathrm{T}})$, in addition to summing up Eq. (6) for $y^{\mathrm{T}} \in \mathcal{Y}^{\mathrm{T}}$, we add Laplace smoothing coefficient $\lambda$ to ensure the denominator of Eq. (5) being non-zero, with $k_{\mathrm{S}}$ being the size of $\mathcal{Y}^{\mathrm{S}}$:

$$\hat{p}_{\mathrm{BLM}}(Y^{\mathrm{S}} = y^{\mathrm{S}}|X^{\mathrm{T}}) = \frac{\sum_{y^{\mathrm{T}} \in \mathcal{Y}^{\mathrm{T}}} \sum_{i=1}^{n} \mathbb{1}\{y_i^{\mathrm{T}} = y^{\mathrm{T}}\} \cdot \mathbb{1}\{\hat{y}_i^{\mathrm{S}} = y^{\mathrm{S}}\} + \lambda}{n + k_{\mathrm{S}} \cdot \lambda} = \frac{\sum_{i=1}^{n} \mathbb{1}\{\hat{y}_i^{\mathrm{S}} = y^{\mathrm{S}}\} + \lambda}{n + k_{\mathrm{S}} \cdot \lambda}. \quad (7)$$

Substituting Eq. (7) and Eq. (6) back to Eq. (5) yields the estimation of $\hat{\omega}_{y^{\mathrm{S}}, y^{\mathrm{T}}}$ to be $\hat{p}_{\mathrm{BLM}}(Y^{\mathrm{T}} = y^{\mathrm{T}}|Y^{\mathrm{S}} = y^{\mathrm{S}}, X^{\mathrm{T}})$. After column-wise sum normalization of $\hat{\omega}_{y^{\mathrm{S}}, y^{\mathrm{T}}}$ to satisfy $\sum_{j=1}^{k_{\mathrm{S}}} \omega_{j,\cdot} = 1$ (as formulated in Section 3), we obtain the final probabilistic LM, denoted as $\omega_{\mathrm{BLM}}$.

**BLM+.** Recall that BLM estimates $p(Y^{\mathrm{T}} = y^{\mathrm{T}}, Y^{\mathrm{S}} = y^{\mathrm{S}}|X^{\mathrm{T}})$ by frequency-counting based on a single *most likely* predicted label. However, this strategy disregards other high-ranking predictions that could offer valuable information. Thus, we introduce BLM+, an extension of BLM that considers top-$K$ *predicted probabilities* of the pretrained model for the estimation of $p(Y^{\mathrm{T}} = y^{\mathrm{T}}, Y^{\mathrm{S}} = y^{\mathrm{S}}|X^{\mathrm{T}})$. Rather than relying solely on the tally, BLM+ aggregates *probabilities* for samples where $y^{\mathrm{S}}$ ranks among the top-$K$ predictions. In this way, BLM+ acknowledges the uncertainty in $f(x_i^{\mathrm{T}})$ and exploits other potential predictions, providing more robust estimations.

Let $\mathcal{Y}_{K,i}^{\mathrm{S}} \equiv \{y' | \arg\max_{y_1, \ldots y_K} f(x_i^{\mathrm{T}})_{y'}\}$ denote the set of the top-$K$ predicted pretrained labels for input $x_i^{\mathrm{T}}$, and $\hat{p}(y^{\mathrm{S}}|x_i^{\mathrm{T}}) \equiv (\mathrm{softmax} \circ f)(x_i^{\mathrm{T}})_{y^{\mathrm{S}}}$ denote the predicted probability for any $y^{\mathrm{S}} \in \mathcal{Y}^{\mathrm{S}}$ given $x_i^{\mathrm{T}}$. Then, within the BLM+ strategy, the joint density is approximated[3] as:

$$\hat{p}_{\mathrm{BLM}_+}(Y^{\mathrm{T}} = y^{\mathrm{T}}, Y^{\mathrm{S}} = y^{\mathrm{S}}|X^{\mathrm{T}}) = \frac{\sum_{i=1}^{n} \mathbb{1}\{y_i^{\mathrm{T}} = y^{\mathrm{T}}\} \cdot \hat{p}(y^{\mathrm{S}}|x_i^{\mathrm{T}}) \cdot \mathbb{1}\{y^{\mathrm{S}} \in \mathcal{Y}_{K,i}^{\mathrm{S}}\}}{n}. \quad (8)$$

Similar to BLM, with the Laplace smoothing coefficient being $\lambda$ and the size of $\mathcal{Y}^{\mathrm{S}}$ being $k_{\mathrm{S}}$, $p(Y^{\mathrm{S}} = y^{\mathrm{S}}|X^{\mathrm{T}})$ can be expressed by applying BLM+ as:

$$\hat{p}_{\mathrm{BLM}_+}(Y^{\mathrm{S}} = y^{\mathrm{S}}|X^{\mathrm{T}}) = \frac{\sum_{i=1}^{n} \hat{p}(y^{\mathrm{S}}|x_i^{\mathrm{T}}) \cdot \mathbb{1}\{y^{\mathrm{S}} \in \mathcal{Y}_{K,i}^{\mathrm{S}}\} + \lambda}{n + k^{\mathrm{S}} \cdot \lambda}. \quad (9)$$

Combining Eq. (9) and Eq. (8) with Eq. (5), and going through all $y^{\mathrm{T}} \in \mathcal{Y}^{\mathrm{T}}$ and $y^{\mathrm{S}} \in \mathcal{Y}^{\mathrm{S}}$, we obtain the full BLM+ estimation as $\omega_{\mathrm{BLM}_+}$ after column-wise sum normalization of $\hat{\omega}_{y^{\mathrm{S}}, y^{\mathrm{T}}}$, similar to BLM. In practice, we set $K = \lfloor \alpha \cdot k_{\mathrm{T}} \rfloor$, with ratio $\alpha$ being a hyper-parameter that decides $K$ based on the size of downstream label space $k_{\mathrm{T}}$.

**Pipeline and Learning Strategy.** The learning of BLM and BLM+ allows for seamless integration into existing VR pipelines. It is model-agnostic (e.g., pretrained ResNet or ResNeXt) and compatible with all input VR methods (e.g., watermarking or padding). Figure 2 illustrates the learning strategy in detail. Besides, the learning pipeline of BLM is shown in Algorithm 1, while that of BLM+ is shown in Algorithm 2. The completed pseudocode for all LM methods (RLM, FLM, ILM, BLM, BLM+) and a more detailed discussion of involved matrix operations are in Appendix D.

---

[3]Note that this approximation is not normalized, and thus, is not strictly equivalent to the true probability.

| **Algorithm 1** Training Pipeline of BLM | **Algorithm 2** Training Pipeline of BLM+ |
|---|---|
| 1: **Input:** Pretrained label space $\mathcal{Y}^{\mathrm{S}}$ with $k_{\mathrm{S}}$ labels, downstream label space $\mathcal{Y}^{\mathrm{T}}$ with $k_{\mathrm{T}}$ labels, downstream training set $\{(x_i^{\mathrm{T}}, y_i^{\mathrm{T}})\}_{i=1}^n$, pretrained model $f_{\mathrm{pre}}(\cdot)$, iterations $E$, learning rate $a$, hyper-parameter $\lambda$ | 1: **Input:** Pretrained label space $\mathcal{Y}^{\mathrm{S}}$ with $k_{\mathrm{S}}$ labels, downstream label space $\mathcal{Y}^{\mathrm{T}}$ with $k_{\mathrm{T}}$ labels, downstream training set $\{(x_i^{\mathrm{T}}, y_i^{\mathrm{T}})\}_{i=1}^n$, pretrained model $f_{\mathrm{pre}}(\cdot)$, iterations $E$, learning rate $a$, $\lambda$, $K$ |
| 2: **Output:** Probabilistic LM $\omega_{\mathrm{BLM}} \in [0,1]^{k_{\mathrm{S}} \times k_{\mathrm{T}}}$ | 2: **Output:** Probabilistic LM $\omega_{\mathrm{BLM+}} \in [0,1]^{k_{\mathrm{S}} \times k_{\mathrm{T}}}$ |
| 3: Initialize $\omega_{\mathrm{BLM}} \leftarrow \{0\}^{k_{\mathrm{S}} \times k_{\mathrm{T}}}$, set $\theta \leftarrow \mathbf{0}$ | 3: Initialize $\omega_{\mathrm{BLM+}} \leftarrow \{0\}^{k_{\mathrm{S}} \times k_{\mathrm{T}}}$, set $\theta \leftarrow \mathbf{0}$ |
| 4: **for** $e = 1...E$ **do** | 4: **for** $e = 1...E$ **do** |
| 5:    # Step 1: Get Pretrained Model Outputs | 5:    # Step 1: Get Pretrained Model Outputs |
| 6:    $f(x_i^{\mathrm{T}}; \theta) = f_{\mathrm{pre}}(f_{\mathrm{in}}(x_i^{\mathrm{T}}; \theta))$ for $i = 1...n$ | 6:    $f(x_i^{\mathrm{T}}; \theta) = f_{\mathrm{pre}}(f_{\mathrm{in}}(x_i^{\mathrm{T}}; \theta))$ for $i = 1...n$ |
| 7:    # Step 2: Compute (or Update) the LM Matrix | 7:    # Step 2: Compute (or Update) the LM Matrix |
| 8:    $\hat{y}_i^{\mathrm{S}} \leftarrow \mathrm{argmax}_{y \in \mathcal{Y}^{\mathrm{S}}} f(x_i^{\mathrm{T}}; \theta)_y$ for $i = 1...n$ | 8:    $\mathcal{Y}_{K,i}^{\mathrm{S}} \leftarrow \{y' | \mathrm{argmax}_{y_1,...,y_K} f(x_i^{\mathrm{T}}; \theta)_{y'}\}$ for $i = 1...n$ |
| 9:    **if** e=1 **then** Compute $\omega_{\mathrm{BLM}}$ using Eq. (5,6,7) | 9:    $\hat{p}(y | x_i^{\mathrm{T}}) \leftarrow \mathrm{softmax}(f(x_i^{\mathrm{T}}; \theta))_y$ for $y \in \mathcal{Y}^{\mathrm{S}}$, $i = 1...n$ |
| 10:   **else** Update $\omega_{\mathrm{BLM}}$ using Eq. (5,6,7) | 10:   **if** e=1 **then** Compute $\omega_{\mathrm{BLM+}}$ using Eq. (5,8,9) |
| 11:   # Step 3: Predict Downstream Labels | 11:   **else** Update $\omega_{\mathrm{BLM+}}$ using Eq. (5,8,9) |
| 12:   $\hat{y}_i^{\mathrm{T}} \leftarrow \mathrm{argmax}_y f_{\mathrm{out}}^\omega(f(x_i^{\mathrm{T}}; \theta))_y$ for $i = 1...n$ | 12:   # Step 3: Predict Downstream Labels |
| 13:   # Step 4: Update VR Patterns | 13:   $\hat{y}_i^{\mathrm{T}} \leftarrow \mathrm{argmax}_y f_{\mathrm{out}}^\omega(f(x_i^{\mathrm{T}}; \theta))_y$ for $i = 1...n$ |
| 14:   $\theta \leftarrow \theta - a \bigtriangledown_\theta \sum_{i=1}^n \ell(y_i^{\mathrm{T}}, f_{\mathrm{out}}^\omega(f(x_i^{\mathrm{T}}; \theta)))$ | 14:   # Step 4: Update VR Patterns |
| 15: **end for** | 15:   $\theta \leftarrow \theta - a \bigtriangledown_\theta \sum_{i=1}^n \ell(y_i^{\mathrm{T}}, f_{\mathrm{out}}^\omega(f(x_i^{\mathrm{T}}; \theta)))$ |
| 16: **return** $\omega_{\mathrm{BLM}}$ | 16: **end for** |
| | 17: **return** $\omega_{\mathrm{BLM+}}$ |

The iterative process of learning $\omega_{\mathrm{BLM}}, \omega_{\mathrm{BLM+}}$ comprises these four steps: 1) Input images, with VR patterns, are fed into the fixed pretrained model to obtain output logits and predicted pretrained labels. 2) BLM and BLM+ replace previous LM (e.g., RLM, FLM or ILM) to estimate $\omega$. 3) The initial logits are reweighted using $\omega_{\mathrm{BLM}}$ or $\omega_{\mathrm{BLM_+}}$, yielding refined predictions for downstream labels. 4) Loss functions (e.g., cross-entropy) and backpropagation are employed to update the input VR.

## 4.2 Theoretical Analysis

Furthermore, we include a justification of why probabilistic many-to-many LM (e.g., BLM and BLM+) should be favored over deterministic one-to-one LM (e.g., RLM, FLM and ILM). Define the label spaces $\mathcal{Y}^{\mathrm{S}} = \{0, 1\}$ and $\mathcal{Y}^{\mathrm{T}} = \{0, 1\}$ as binary sets[4]. Consider the set of potential LM functions $\mathcal{F}_{\mathrm{lm}} = \{f_{\mathrm{lm}} : \mathcal{Y}^{\mathrm{S}} \to \mathcal{Y}^{\mathrm{T}}\}$, including each function $f_{\mathrm{lm}}(y^{\mathrm{S}}) \in \{y^{\mathrm{T}}, 1 - y^{\mathrm{T}}\}$. For any $f_{\mathrm{lm}} \in \mathcal{F}_{\mathrm{lm}}$, the expected accuracy of $f_{\mathrm{lm}}$ regarding the entire downstream label space is defined as[5]:

$$\mathrm{Acc}(f_{\mathrm{lm}}) = \mathbb{E}_{y^{\mathrm{T}} \in \mathcal{Y}^{\mathrm{T}}} \left[ \sum_{y^{\mathrm{S}} \in \mathcal{Y}^{\mathrm{S}}} p(y^{\mathrm{S}}) \cdot p\left(f_{\mathrm{lm}}(y^{\mathrm{S}}) = y^{\mathrm{T}} | y^{\mathrm{S}}\right) \right], \tag{10}$$

where $p(y^{\mathrm{S}})$ is the marginal distribution of the pretrained labels and $p\left(f_{\mathrm{lm}}(y^{\mathrm{S}}) = y^{\mathrm{T}} | y^{\mathrm{S}}\right)$ is the conditional probability that $f_{\mathrm{lm}}$ correctly predicts a downstream label $y^{\mathrm{T}}$ from a pretrained label $y^{\mathrm{S}}$. Let $f_{\mathrm{plm}}$ and $f_{\mathrm{dlm}}$ denote the probabilistic LM (Definition E.1) and deterministic LM (Definition E.2), respectively. We finally prove that $\mathrm{Acc}(f_{\mathrm{plm}}) \geq \mathrm{Acc}(f_{\mathrm{dlm}})$ (Corollary E.5) in Appendix E, which further verifies the effectiveness of our methods in the view of theoretical understanding.

## 5 Experiments

**Tasks and Baselines.** Following ILM [4], we employ ResNet-18 [17] pretrained on ImageNet-1K [46] and ResNeXt pretrained on Instagram [37] to test the performance of VR. The results are evaluated on twelve downstream datasets: Flowers102 [41], DTD [9], UCF101 [47], Food101 [2], GTSRB [19], EuroSAT [18], OxfordPets [44], StanfordCars [29], SUN397 [58], CIFAR10/100 [30] and SVHN [39]. Previous gradient-free LM methods RLM [12], FLM [48] and ILM [4] are used as the baselines. The results of deep learning-based LM will also be included for reference, where LM is treated as a single-layer linear neural network connected to the output of the pretrained model

---

[4]This analysis focuses on the binary setting for simplicity.

[5]Input $x$ is intentionally omitted as all LM methods operate on the same inputs.

Table 1: Performance comparison of gradient-free output LM methods (mean % ± std %). Ours are highlighted and the highest accuracy is in **bold** (with deep learning-based LM in gray for reference)

| | ResNet-18 (ImageNet-1K) | | | | | | ResNeXt-101-32x8d (Instagram) | | | | |
| --- | --- | --- | --- | --- | --- | --- | --- | --- | --- | --- | --- |
| Padding | Gradient-free | | | | | Deep | Gradient-free | | | | Deep |
| Methods | RLM | FLM | ILM | BLM | BLM+ | - | FLM | ILM | BLM | BLM+ | - |
| Flowers102 | $11.0_{\pm0.5}$ | $20.0_{\pm0.3}$ | $27.9_{\pm0.7}$ | $44.4_{\pm1.1}$ | $\mathbf{50.1}_{\pm0.6}$ | $76.7_{\pm0.2}$ | $22.5_{\pm0.5}$ | $27.9_{\pm0.3}$ | $\mathbf{31.5}_{\pm0.4}$ | $30.1_{\pm0.7}$ | $85.2_{\pm1.3}$ |
| DTD | $16.3_{\pm0.7}$ | $32.4_{\pm0.5}$ | $35.3_{\pm0.9}$ | $42.0_{\pm0.5}$ | $\mathbf{43.9}_{\pm0.4}$ | $49.1_{\pm0.3}$ | $40.3_{\pm0.5}$ | $41.4_{\pm0.7}$ | $47.8_{\pm0.4}$ | $\mathbf{49.4}_{\pm0.4}$ | $64.0_{\pm0.2}$ |
| UCF101 | $6.6_{\pm0.4}$ | $18.9_{\pm0.5}$ | $23.9_{\pm0.5}$ | $30.9_{\pm1.1}$ | $\mathbf{32.0}_{\pm0.4}$ | $46.0_{\pm0.6}$ | $41.9_{\pm0.6}$ | $43.1_{\pm0.8}$ | $48.3_{\pm0.1}$ | $\mathbf{50.1}_{\pm0.6}$ | $68.3_{\pm0.1}$ |
| Food101 | $3.8_{\pm0.3}$ | $12.8_{\pm0.1}$ | $14.8_{\pm0.2}$ | $23.2_{\pm0.1}$ | $\mathbf{25.1}_{\pm0.3}$ | $34.1_{\pm0.1}$ | $20.5_{\pm0.5}$ | $23.0_{\pm0.4}$ | $29.6_{\pm0.6}$ | $\mathbf{31.4}_{\pm0.2}$ | $58.7_{\pm0.3}$ |
| GTSRB | $46.1_{\pm1.3}$ | $45.5_{\pm1.0}$ | $52.0_{\pm1.2}$ | $\mathbf{54.8}_{\pm0.7}$ | $54.3_{\pm0.7}$ | $63.1_{\pm0.5}$ | $56.2_{\pm0.6}$ | $59.9_{\pm1.0}$ | $62.9_{\pm0.5}$ | $\mathbf{63.0}_{\pm0.8}$ | $74.4_{\pm0.5}$ |
| EuroSAT | $82.4_{\pm0.4}$ | $83.8_{\pm0.2}$ | $85.2_{\pm0.6}$ | $\mathbf{86.7}_{\pm0.2}$ | $\mathbf{86.7}_{\pm0.1}$ | $92.4_{\pm0.1}$ | $87.8_{\pm0.4}$ | $86.2_{\pm0.8}$ | $87.6_{\pm0.3}$ | $\mathbf{88.3}_{\pm0.3}$ | $93.2_{\pm0.1}$ |
| OxfordPets | $9.3_{\pm0.4}$ | $62.9_{\pm0.1}$ | $65.4_{\pm0.7}$ | $69.8_{\pm0.3}$ | $\mathbf{70.6}_{\pm0.2}$ | $73.0_{\pm0.3}$ | $76.8_{\pm0.6}$ | $78.9_{\pm0.8}$ | $82.4_{\pm0.4}$ | $\mathbf{83.0}_{\pm0.6}$ | $91.8_{\pm0.1}$ |
| StanfordCars | $0.9_{\pm0.1}$ | $2.7_{\pm0.1}$ | $4.5_{\pm0.1}$ | $5.4_{\pm0.1}$ | $\mathbf{7.7}_{\pm0.1}$ | $14.3_{\pm0.1}$ | $4.6_{\pm0.1}$ | $7.0_{\pm0.2}$ | $8.3_{\pm0.1}$ | $\mathbf{9.3}_{\pm0.3}$ | $50.5_{\pm0.5}$ |
| SUN397 | $1.0_{\pm0.1}$ | $10.4_{\pm0.1}$ | $13.0_{\pm0.2}$ | $16.2_{\pm0.1}$ | $\mathbf{18.7}_{\pm0.3}$ | $26.3_{\pm0.6}$ | $21.6_{\pm0.3}$ | $23.7_{\pm0.2}$ | $30.1_{\pm0.1}$ | $\mathbf{32.0}_{\pm0.3}$ | $51.5_{\pm0.8}$ |
| CIFAR10 | $63_{\pm0.1}$ | $65.7_{\pm0.6}$ | $65.5_{\pm0.1}$ | $66.7_{\pm0.2}$ | $\mathbf{66.8}_{\pm0.2}$ | $72.1_{\pm0.8}$ | $80.3_{\pm0.3}$ | $81.7_{\pm0.3}$ | $\mathbf{82.2}_{\pm0.3}$ | $\mathbf{82.2}_{\pm0.1}$ | $83.4_{\pm0.1}$ |
| CIFAR100 | $12.9_{\pm0.1}$ | $18.1_{\pm0.2}$ | $24.8_{\pm0.1}$ | $29.6_{\pm0.6}$ | $\mathbf{30.6}_{\pm0.4}$ | $46.7_{\pm0.2}$ | $39.7_{\pm0.2}$ | $45.9_{\pm0.2}$ | $\mathbf{47.8}_{\pm0.3}$ | $\mathbf{47.8}_{\pm0.3}$ | $56.2_{\pm0.4}$ |
| SVHN | $73.5_{\pm0.3}$ | $73.1_{\pm0.2}$ | $\mathbf{75.2}_{\pm0.2}$ | $74.5_{\pm0.7}$ | $74.2_{\pm0.3}$ | $82.1_{\pm0.2}$ | $79.0_{\pm0.5}$ | $\mathbf{81.4}_{\pm0.1}$ | $79.8_{\pm0.3}$ | $79.3_{\pm0.4}$ | $85.7_{\pm0.2}$ |
| Average | 27.2 | 37.2 | 40.6 | 45.3 | **46.7** | 56.3 | 47.6 | 50.0 | 53.2 | **53.8** | 71.9 |

for training alongside VR. More dataset and implementation details are in Appendix F. Regarding hyper-parameters of BLM, $\lambda$ is set as 1, and the top-$K$ ratio $\alpha$ is 0.15 (analyzed in Appendix G).

**Results for Padding-based VR.** Padding-based input VR adds trainable noise to the outer frames of centered images. Table 1 shows the performance of BLM and BLM+ applied with padding-based input VR. BLM and BLM+ yield the highest accuracy across all datasets except for SVHN. On ResNet-18, compared to the SOTA (i.e., ILM), BLM achieves an average improvement of 4.7% across the 12 datasets, whereas BLM+ achieves a 6.1% enhancement. On ResNeXt-101, BLM and BLM+ achieve accuracy improvements of 3.2% and 3.8% on average, respectively. The elevation in accuracy is particularly pronounced in tasks with a higher number of classes (e.g., UCF101, CIFAR100). On SVHN, ILM performs slightly better, which could be attributed to the minimal inter-class variation and the smaller number of classes (which is 10) in SVHN, resulting in similar mapping values for different downstream labels and thus reducing our method's advantage (discussed in Appendix H). However, compared to current gradient-free LM methods, the deep learning-based LM may still have an advantage in the performance of downstream tasks due to the learning capacity of the linear layer neural network. Our proposed BLM and BLM+ aim to bridge the gap between gradient-free LM and deep learning-based LM. Additionally, BLM and BLM+ have been observed to possess greater interpretability (see Appendix I for more experiments) and fewer parameters (see Appendix J for details) compared to deep learning-based LM.

Table 2: Performance comparison of gradient-free LM methods for watermarking-based VR on ResNet-18 (mean % ± std %). Ours are highlighted and the highest accuracy is in **bold** ( with deep learning-based LM in gray for reference)

| Watermarking | Gradient-free | | | Deep |
| --- | --- | --- | --- | --- |
| Methods | ILM | BLM | BLM+ | - |
| Flowers102 | $23.2_{\pm0.5}$ | $39.2_{\pm0.6}$ | $\mathbf{44.1}_{\pm0.9}$ | $82.4_{\pm0.4}$ |
| DTD | $29.0_{\pm0.7}$ | $40.1_{\pm0.2}$ | $\mathbf{43.0}_{\pm0.2}$ | $48.9_{\pm0.5}$ |
| UCF101 | $24.4_{\pm0.9}$ | $32.9_{\pm0.8}$ | $\mathbf{35.4}_{\pm0.5}$ | $53.1_{\pm0.2}$ |
| Food101 | $13.2_{\pm0.1}$ | $21.5_{\pm0.4}$ | $\mathbf{22.9}_{\pm0.1}$ | $30.4_{\pm0.9}$ |
| GTSRB | $76.8_{\pm0.9}$ | $\mathbf{82.1}_{\pm0.7}$ | $82.0_{\pm0.8}$ | $89.5_{\pm0.3}$ |
| EuroSAT | $84.3_{\pm0.5}$ | $84.4_{\pm0.5}$ | $\mathbf{84.8}_{\pm0.2}$ | $89.2_{\pm0.2}$ |
| OxfordPets | $70.0_{\pm0.6}$ | $72.4_{\pm0.6}$ | $\mathbf{73.3}_{\pm0.1}$ | $77.6_{\pm0.8}$ |
| StanfordCars | $3.4_{\pm0.1}$ | $5.5_{\pm0.1}$ | $\mathbf{7.4}_{\pm0.1}$ | $30.7_{\pm0.3}$ |
| SUN397 | $13.4_{\pm0.2}$ | $18.4_{\pm0.1}$ | $\mathbf{19.4}_{\pm0.2}$ | $32.9_{\pm0.3}$ |
| CIFAR10 | $68.9_{\pm0.4}$ | $74.9_{\pm0.2}$ | $\mathbf{75.7}_{\pm0.1}$ | $71.7_{\pm0.6}$ |
| CIFAR100 | $33.8_{\pm0.2}$ | $41.2_{\pm0.3}$ | $\mathbf{41.6}_{\pm0.3}$ | $39.9_{\pm0.5}$ |
| SVHN | $78.3_{\pm0.3}$ | $\mathbf{79.2}_{\pm0.1}$ | $78.8_{\pm0.2}$ | $83.7_{\pm0.2}$ |
| Average | 43.2 | 49.3 | **50.7** | 60.8 |

**Results for Watermarking-based VR.** BLM and BLM+ can be applied to different input VR methods. For the watermarking-based VR method, which overlays trainable noise patterns on resized images, the results of BLM and BLM+ with ResNet-18 as the pretrained model are shown in Table 2. Since ILM is the best-performing baseline, we only include its results here for comparison. Our BLM and BLM+ methods again outperform ILM, achieving an average gain in accuracy of 6.1% and 7.5%, respectively. Therefore, in the case of watermarking-based VR, BLM and BLM+ also close the gap between current gradient-free and deep learning-based LM. Results in Tabel 2 underscore the applicability of our output LM methods with different input VR.

**Results for Vision-Language Models.** The application of our BLM and BLM+ on vision-language models (i.e., CLIP), along with the performance, and visualization results are dis-

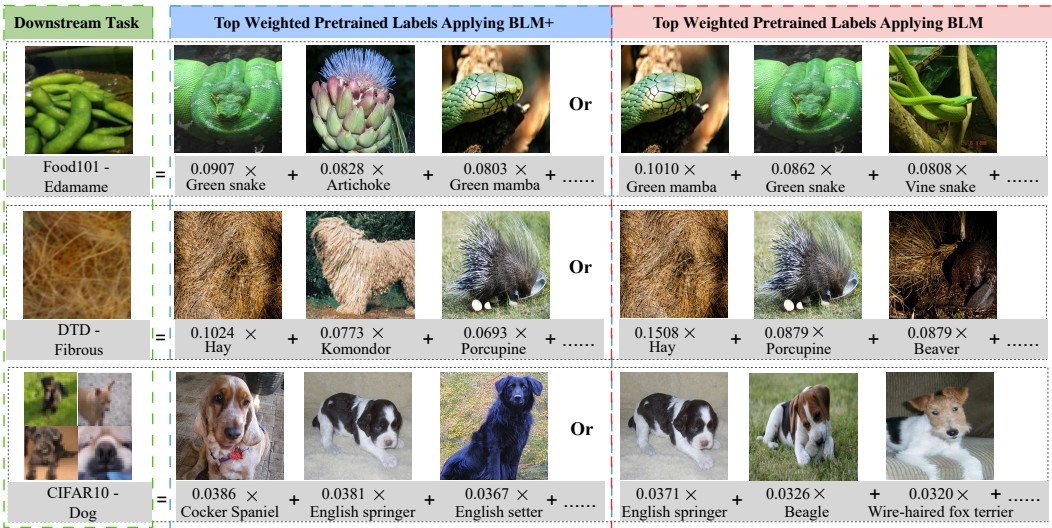

Figure 3: Visualization results of top weighted pretrained labels $y^S$ and weights $\omega_{y^S, y^T}$ for some $y^T$ applying BLM and BLM+. Downstream labels 'Edamame', 'Fibrous', and 'Dog' are shown as examples. ResNet-18 pretrained on ImageNet is used. More results are in Appendix K.

cussed in Appendix L. BLM and BLM+ achieve the average accuracy of 79.1% and 79.3% across 12 datasets, respectively, and outperform the baseline methods on 11 datasets.

**Ablation Study.** Table 3 presents the ablation study results of BLM and BLM+. For BLM, we list the results of replacing probabilistic LM with a one-to-one LM, denoted as '-Bayes', and the results of calculating $\omega_{BLM}$ only once in the first epoch without subsequent iterations, denoted as '-Iter'. For BLM+, the removal of aggregating probabilities results in BLM; hence, we report the results of aggregating all probabilities instead of top-$K$ predicted probabilities, denoted as '-Top-K'. Like that for BLM, '-Iter' shows the results of calculating $\omega_{BLM+}$ only once without subsequent iterations. Besides, when both '-Top-K' and '-Bayes' are applied to BLM+, BLM+ degenerates into the same results as '-Bayes' of BLM, which is displayed in the previous column of Table 3.

Table 3: Ablation study results of BLM and BLM+, using ResNet-18 as the pretrained model (showing the mean accuracies (%), with ours highlighted and the best in **bold**)

| Method | BLM | | | BLM+ | | |
|---|---|---|---|---|---|---|
| | - Bayes | - Iter | Ours | - Top-K | - Iter | Ours |
| Flowers102 | 27.9 | 30.8 | **44.4** | 48.2 | 43.6 | **50.1** |
| DTD | 35.3 | 38.0 | **42.0** | 42.4 | 42.0 | **43.9** |
| UCF101 | 23.9 | 28.8 | **30.9** | 30.9 | **33.2** | 32.0 |
| Food101 | 14.8 | 18.4 | **23.2** | 23.6 | 22.8 | **25.1** |
| GTSRB | 52.0 | 44.6 | **54.8** | 50.5 | 44.8 | **54.3** |
| EuroSAT | 85.2 | 85.0 | **86.7** | 85.1 | 85.4 | **86.7** |
| OxfordPets | 65.4 | 68.1 | **69.8** | 59.9 | 70.6 | **70.6** |
| StanfordCars | 4.5 | 2.8 | **5.4** | 6.5 | 6.2 | **7.7** |
| SUN397 | 13.0 | 14.5 | **16.2** | 17.3 | 16.4 | **18.7** |
| CIFAR10 | 65.5 | 63.9 | **66.7** | 65.2 | 63.5 | **66.8** |
| CIFAR100 | 24.8 | 23.2 | **29.6** | **30.8** | 26.2 | 30.6 |
| SVHN | **75.2** | 64.6 | 74.5 | 70.0 | 62.6 | **74.2** |

For BLM, employing 'Bayes' to compute $\omega$ improves accuracy across most datasets, with slightly smaller gains observed for datasets with fewer classes (EuroSAT, CIFAR10, and SVHN). For BLM+, applying 'Top-K' assists in filtering out redundant information, yielding positive impacts across most datasets. In particular, on the OxfordPets dataset, to classify cat and dog breeds, using top-$K$ predicted probability effectively filters out numerous irrelevant categories in the pretrained label space, which leads to significant improvements. Furthermore, for both BLM and BLM+, iterative updates are crucial as the initial input VR may deviate considerably from the final iteration. The greater the disparity between the domains of downstream and pretrained tasks (GTSRB, SVHN), the more pronounced the impact of the input VR, thereby emphasizing the necessity of iteration updates.

**Visualization Results.** The probabilistic LM obtained by BLM or BLM+ can elucidate the connection between pretrained and downstream label spaces. Figure 3 shows the visualization results for three

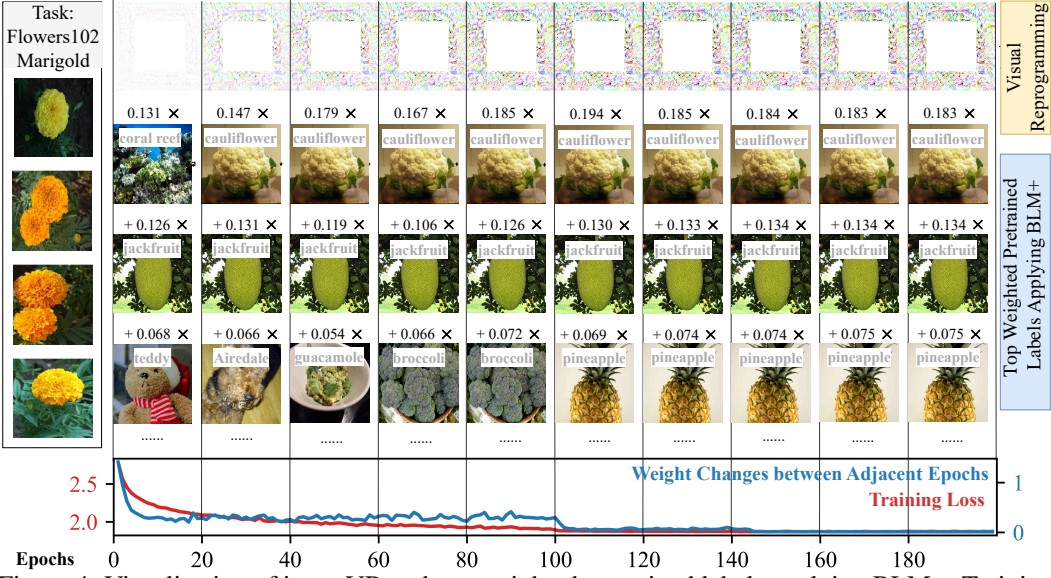

Figure 4: Visualization of input VR and top-weighted pretrained labels applying BLM+. Training loss and weight changes (Euclidean norm) of probabilistic LM $\omega_{\text{BLM+}}$ per iteration are plotted below. Pretrained ResNet-18 is used, and the downstream label 'Marigold' is selected as an example.

labels in downstream tasks, taking ResNet-18 pretrained on ImageNet-1K as an example. Each column of $\omega$ computed using BLM or BLM+ is a vector with length $k_{\text{S}} = 1000$, representing the weights assigned to the 1,000 outputs–one for each $y^{\text{S}}$–of the pretrained model corresponding to a downstream label $y^{\text{T}}$. The top-weighted labels (i.e., $y^{\text{S}}$ where $\omega_{y^{\text{S}}, y^{\text{T}}}$ is larger) for 'Edamame' correspond to organisms such as snakes and artichokes, which share similarities in color and shape. Similarly, the predominant labels associated with 'Fibrous' from the texture dataset include rough-textured items like 'Hay' and 'Komondor'. 'Dog' encompasses various sub-breed canines. These findings suggest that BLM and BLM+ establish an optimal probabilistic LM between label spaces, and handle similarity or inclusion relationship, addressing the drawbacks in Figure 1.

**Discussion of Why VR Is Effective.** From a visualization perspective, Figure 4 shows the top-weighted pretrained labels and input VR patterns $\theta$ at different iteration stages using BLM+. The training loss for each iteration and changes in $\omega$, measured by the Euclidean norm, are also plotted. During the update of $\omega$ and $\theta$, the pretrained labels with top $\omega_{y^{\text{S}}, y^{\text{T}}}$ for $y^{\text{T}}$ being 'Marigold' transition gradually from dissimilar labels such as 'Reef' and 'Teddy' to 'Cauliflower' and 'Pineapple' which share more similarities in color, shape and texture. Meanwhile, the training loss diminishes gradually, and $\omega$ converges, demonstrating the effectiveness of VR and BLM+.

**Impact of Label Space Sizes $k_{\text{T}}$.** Figure 5 shows the relationship between different sizes of the downstream label space and the accuracy improvement achieved by BLM and BLM+. Tasks with larger label spaces report more pronounced performance improvements. While simpler tasks with smaller label spaces might not fully showcase the power of our approach, the strength of BLM and BLM+ lies in unraveling the complex many-to-many relationship that often arises in tasks with more numerous classes. In such scenarios, our probabilistic LM methods demonstrate their full potential.

**Impact of Training Dataset Sizes $n$.** Figure 6 illustrates the impact of varying training dataset sizes for the downstream task on different LM methods. Regarding CIFAR100 as the downstream task, compared with RLM and ILM, BLM and BLM+ yield higher accuracy consistently. With approximately a 40% fraction of the downstream training data, BLM or BLM+ can achieve similar accuracy compared with training on the entire dataset.

**Other Experiments.** The parameter experiments and performance analysis regarding the impact of Laplace smoothing coefficient $\lambda$ and top-$K$ ratio $\alpha$ for BLM and BLM+ are detailed in Appendix G. The visualization and analysis of LM matrices derived from gradient-free and deep learning-based methods can be found in Appendix I. Training cost analysis is discussed in Appendix J. Additional visualization results of LM methods applied to pretrained vision models are presented in Appendix K. Lastly, the application of BLM and BLM+ for vision-language models is explored in Appendix L.

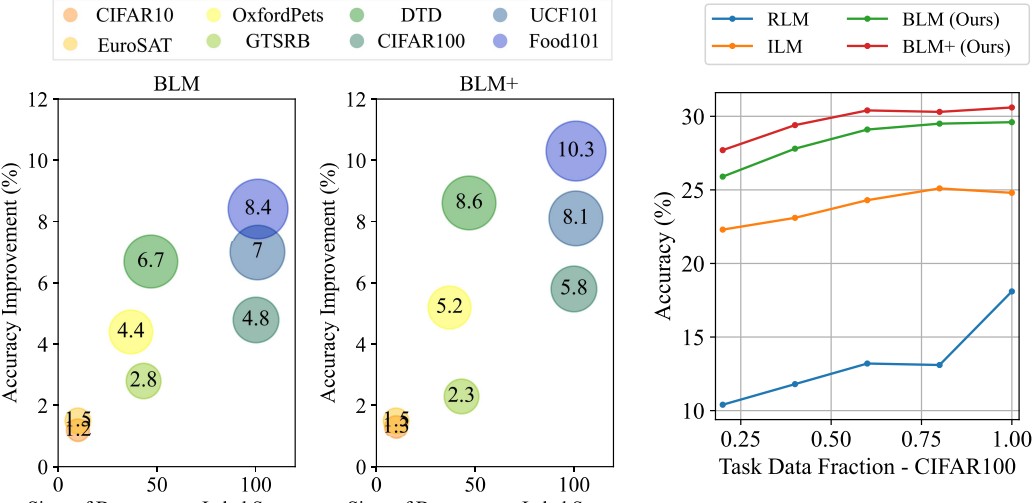

Figure 5: Accuracy improvement (%) of BLM and BLM+ compared with ILM given different sizes ($k_T$) of the downstream label space, using pretrained ResNet-18.

Figure 6: Accuracy (%) of methods when varying training dataset sizes $n$ for downstream task CIFAR100, using pretrained ResNet-18.

## 6 Conclusion

We focus on output LM methods for VR and reveal the drawbacks in current gradient-free LM methods, which use one-to-one mappings that overly simplify the relationship between the pretrained and downstream label spaces. To address this issue, we propose BLM, which calculates probabilistic LM matrices guided by Bayes' theorem. Additionally, we aggregate the probability of top-$K$ predicted pretrained labels instead of counting a single label during the estimation of probabilistic LM matrices, yielding an improved method BLM+. Both theoretical analysis and experimental results validate the effectiveness of BLM and BLM+ while offering insights into understanding the effectiveness of VR through a probabilistic lens.

## Acknowledgement

CYC, ZSY, and FL are supported by the Australian Research Council (ARC) with grant number DE240101089, and FL is also supported by ARC with grant number DP230101540 and the NSF&CSIRO Responsible AI program with grant number 2303037. JZQ is supported by ARC with grant number DP240101006. This research is also supported by The University of Melbourne's Research Computing Services and the Petascale Campus Initiative. We sincerely appreciate the time and dedication of the reviewers in carefully reviewing our manuscript.

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

# A   The Problem Setting of VR

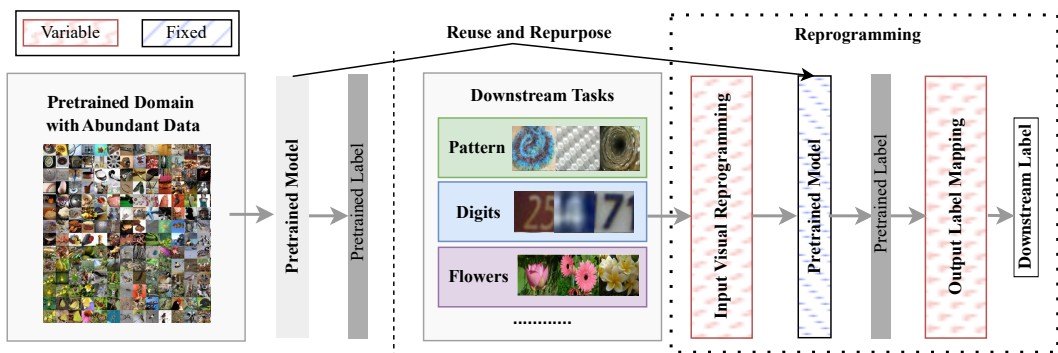

Figure 7: The problem setting of Visual Reprogramming. The left part shows the pretrained model and corresponding dataset, while the right part shows downstream tasks. The pretrained model is fixed, whereas the input VR and output LM modules are variable.

The task of VR reuses fixed pretrained models for downstream tasks. As illustrated in Figure 7, an input VR module operates before pretrained models, directly altering the input space of downstream tasks. Concurrently, an output LM function acts after pretrained models, taking the predicted pretrained labels as input and outputting those for downstream tasks. Hence, VR achieves the reusability of pretrained models for downstream tasks without adapting the model parameters, primarily through modifications to the input and output spaces.

# B   Recent Work in Transfer Learning

Visual reprogramming is one type of methods that aim to obtain models on downstream tasks with the help of pretrained models. This process is similar to the aim of transfer learning which is used to leverage knowledge from a data-rich domain [13] or a pretrained model [51] to address tasks on other domains. The former is known as *domain adaptation*, and the latter is known as finetuning.

**Finetuning.** Given a pretrained model, finetuning uses trainable parameters to accommodate new task-specific information of the downstream tasks. As pretrained models grow in size, recent progresses in transfer learning have prioritized *parameter-efficient finetuning* (PEFT) [20] to support resource-friendly adaptation. Regarding PEFT, the prevailing methods can be categorized as follows. The most widely adopted approach is selective finetuning [46, 63], which adjusts a subset of parameters from the pretrained model while keeping the remaining components fixed, thereby reducing the total number of trainable parameters for downstream tasks. Other methods may involve adding adapters [5, 31, 43], which introduce extra trainable layers or parameters to the pretrained model and finetune only these adapters during training. Moreover, low-rank adaptation methods [20, 64, 69] have also been proposed for pretrained Vision Transformers. They apply low-rank decomposition to the parameters during training, achieving remarkable performance on downstream tasks with a significantly reduced number of parameters. Additionally, Prompt Tuning methods [16, 22, 53], similarly directed at pretrained Vision Transformers, integrate trainable parameters parallel to the features at the input and intermediate layers. The primary distinction of these methods from VR [1, 3, 4, 48, 49, 65] lies in their necessity to be designed according to different pretrained model architectures and may also involve modifying the model weights. In contrast, VR is model-agnostic and does not require alterations to pretrained model parameters.

**Domain Adaptation.** *Domain adaptation* (DA) aims to bridge the distributional gap by aligning feature spaces of the source task to the target domain with different data distributions [14, 36, 54]. Often, DA is achieved by learning invariant representations or transforming parameters to manage domain-specific shifts of the source and target data. CDAN [33] addresses this by introducing a conditional discriminator for class label-conditioned feature adaptation, while UDA [34] leverages residual layers to capture both domain-specific and domain-shared representations. More recently, source-free DA [32, 62, 66], which seeks adaptation without access to the source data, has gained

popularity due to growing concerns over data privacy and storage limitations, as well as the need for adaptation in scenarios where source data is inaccessible [8, 10].

**Pretrained Model Selection.** The BLM/BLM+ framework models $p(Y^T|X^T)$ by using the pretrained model $f_{\text{pre}}$ and integrating over the pre-traiend label space $\mathcal{Y}^S$ (Eq. (3)), relying on $n$ training examples for empirical estimation (Eq. (4)). Essentially, Eq. (4) can be seen as a score that assesses how a pre-trained model, whose output lies in $\mathcal{Y}^S$, performs on a downstream task $\mathcal{X}^T \times \mathcal{Y}^T$, measured over $n$ downstream samples. This finding resembles LEEP [40], a pre-trained model transferability metric, which shares the same empirical formulation with BLM. In this sense, BLM/BLM+ extends LEEP by incorporating top-$K$ aggregation for more robust estimation, suggesting the potential as a more generalized framework for assessing pre-trained model transferability.

# C   A Simple Probability Estimation Example

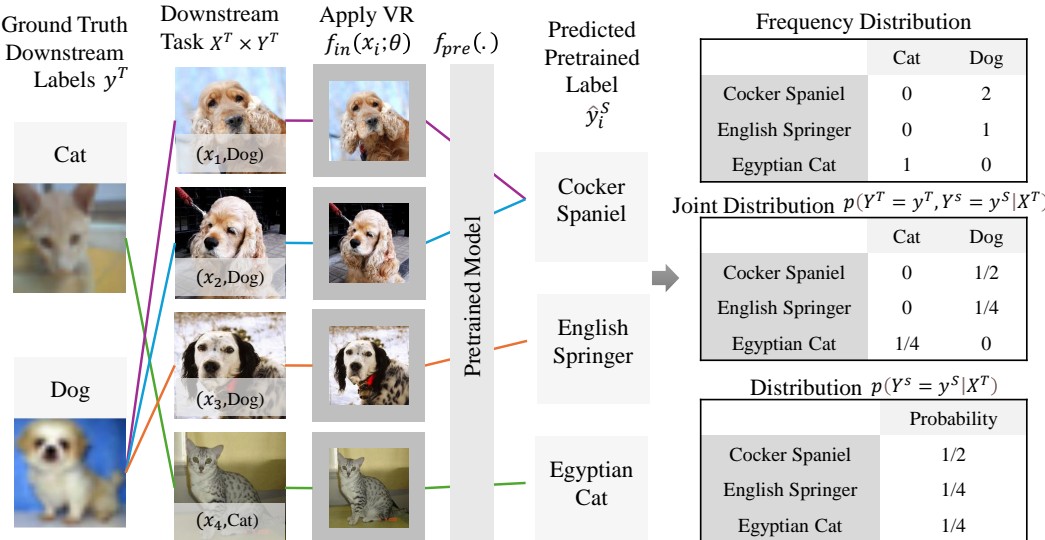

Figure 8: A simple example to help understand how to estimate $p(Y^T = y^T, Y^S = y^S \mid X^T)$ and $p(Y^S = y^S \mid X^T)$.

We aim to estimate $p(Y^T = y^T, Y^S = y^S \mid X^T)$ and $p(Y^S = y^S \mid X^T)$ for BLM and BLM+ in this paper. Here, we employ a simple example (without Laplace smoothing) to help understand how to estimate these two probabilities.

The conditional probability $p(Y^T = y^T, Y^S = y^S \mid X^T)$ represents the joint distribution of $Y^T$ and $Y^S$, given the input reprogramming $f_{\text{in}}(\cdot; \theta)$, the pretrained model $f_{\text{pre}}(\cdot)$, and the variable $X^T$ of the downstream task. Similarly, $p(Y^S = y^S \mid X^T)$ represents the distribution of $Y^S$ under these conditions.

We consider the following example shown in Figure 8. It is assumed that $\mathcal{Y}^T = \{\texttt{Cat}, \texttt{Dog}\}$ and $\mathcal{Y}^S = \{\texttt{CockerSpaniel}, \texttt{EnglishSpringer}, \texttt{EgyptianCat}\}$. The Downstream samples are

$$\{(x_1, \texttt{Dog}), (x_2, \texttt{Dog}), (x_3, \texttt{Dog}), (x_4, \texttt{Cat})\} \in \mathcal{X}^T \times \mathcal{Y}^T.$$

If the reprogrammed predictions calculated by $f_{\text{pre}}(f_{\text{in}}(x_i; \theta))$ are

$$\{x_1 : \texttt{CockerSpaniel}, x_2 : \texttt{CockerSpaniel}, x_3 : \texttt{EnglishSpringer}, x_4 : \texttt{EgyptianCat}\},$$

then the joint distribution $p(Y^{\mathrm{T}} = y^{\mathrm{T}}, Y^{\mathrm{S}} = y^{\mathrm{S}} \mid X^{\mathrm{T}})$ can be estimated as a matrix with the following nonzero values:

$$p(Y^{\mathrm{T}} = \mathtt{Dog}, Y^{\mathrm{S}} = \mathtt{CockerSpaniel} \mid X^{\mathrm{T}}) = \frac{1}{2},$$

$$p(Y^{\mathrm{T}} = \mathtt{Dog}, Y^{\mathrm{S}} = \mathtt{EnglishSpringer} \mid X^{\mathrm{T}}) = \frac{1}{4},$$

$$p(Y^{\mathrm{T}} = \mathtt{Cat}, Y^{\mathrm{S}} = \mathtt{EgyptianCat} \mid X^{\mathrm{T}}) = \frac{1}{4},$$

as is shown in Figure 8. Similarly, $p(Y^{\mathrm{S}} = y^{\mathrm{S}} \mid X^{\mathrm{T}})$ can also be estimated.

## D  Detailed Procedures of Output LM Methods

This section provides a detailed exposition of gradient-free LM methods. Such methods, derived from data distributions, obviate the need for gradients in the output mapping phase. The pseudocode is presented below. Similar to Section 3, $\omega$ represents the one-to-one LM or probabilistic LM.

### D.1  Random Label Mapping (RLM)

---
**Algorithm 3** Random Label Mapping for VR

---
1: **Input:** Pretrained label space $\mathcal{Y}^{\mathrm{S}}$ with $k_{\mathrm{S}}$ labels, downstream label space $\mathcal{Y}^{\mathrm{T}}$ with $k_{\mathrm{T}}$ labels
2: **Output:** One-to-one LM $\omega \in \{0, 1\}^{k_{\mathrm{S}} \times k_{\mathrm{T}}}$
3: Initialize $\omega \leftarrow \{0\}^{k_{\mathrm{S}} \times k_{\mathrm{T}}}$, temp set $T \leftarrow \{\}$ to store matched pretrained labels
4: # Computing output mapping $\omega$
5: **for** $y^{\mathrm{T}} \in \mathcal{Y}^{\mathrm{T}}$ **do**
6:     Randomly select $y^{\mathrm{S}} \in \mathcal{Y}^{\mathrm{S}} - T$
7:     $\omega_{y^{\mathrm{S}}, y^{\mathrm{T}}} \leftarrow 1$
8:     $T \leftarrow T \cup \{y^{\mathrm{S}}\}$
9: **end for**
10: **return** $\omega$

---

The process of *random label mapping* (RLM) is outlined in Algorithm 3. , where the computation of $\omega$ does not involve the downstream training set. The algorithm establishes a random one-to-one mapping between the pretrained and the downstream labels, ensuring that each $y^{\mathrm{T}}$ corresponds to a unique $y^{\mathrm{S}}$. RLM is computed once before learning the input VR $f(\cdot; \theta)$.

### D.2  Frequent Label Mapping (FLM)

---
**Algorithm 4** Computing Frequency Distribution Matrix of [predicted pretrained label, ground-truth downstream label]

---
1: **Input:** Downstream training set $\{(x_i^{\mathrm{T}}, y_i^{\mathrm{T}})\}_{i=1}^n$, given input VR $f_{\mathrm{in}}(\cdot; \theta)$ and pretrained model $f_{\mathrm{pre}}(\cdot)$ with the $j$th dimension being $f_{\mathrm{pre}}(\cdot)_j$
2: **Output:** Frequency distribution matrix $d \in \mathbb{Z}^{k_{\mathrm{S}} \times k_{\mathrm{T}}}$
3: Initialize $d \leftarrow \{0\}^{k_{\mathrm{S}} \times k_{\mathrm{T}}}$
4: # Computing frequency distribution matrix $d$
5: **for** $i = 1...n$ **do**
6:     $\hat{y}_i^{\mathrm{S}} \leftarrow \arg\max_j (f_{\mathrm{pre}}(f_{\mathrm{in}}(x_i^{\mathrm{T}}; \theta))_j)$
7:     $d_{\hat{y}_i^{\mathrm{S}}, y_i^{\mathrm{T}}} \leftarrow d_{\hat{y}_i^{\mathrm{S}}, y_i^{\mathrm{T}}} + 1$
8: **end for**
9: **return** $d$

---

The procedure of *frequent label mapping* (FLM) is outlined in Algorithm 5. Initially, it utilizes the pretrained model to obtain predicted pretrained labels for samples of the downstream task. Subsequently, it computes a joint distribution matrix between the predicted pretrained labels and the ground-truth downstream labels. Finally, employing a greedy algorithm, it iteratively identifies the

---
**Algorithm 5** Frequent Label Mapping for VR
---
1: **Input:** Pretrained label space $\mathcal{Y}^{\mathrm{S}}$ with $k_{\mathrm{S}}$ labels, downstream label space $\mathcal{Y}^{\mathrm{T}}$ with $k_{\mathrm{T}}$ labels, downstream training set $\{(x_i^{\mathrm{T}}, y_i^{\mathrm{T}})\}_{i=1}^n$, given pretrained model $f_{\mathrm{pre}}(\cdot)$
2: **Output:** One-to-one LM $\omega \in \{0,1\}^{k_{\mathrm{S}} \times k_{\mathrm{T}}}$
3: Initialize $\omega \leftarrow \{0\}^{k_{\mathrm{S}} \times k_{\mathrm{T}}}$, temp set $T \leftarrow \{\}$ to store matched pretrained labels, initialize $f_{\mathrm{in}}(\cdot; \theta)$ ($\theta \leftarrow \mathbf{0}$)
4: # Computing frequency distribution matrix $d$
5: Use Algorithm 4 to obtain $d$
6: # Computing output mapping $\omega$
7: **while** size of $T$ is not $k_{\mathrm{T}}$ **do**
8:    Find the maximum $d_{y^{\mathrm{S}}, y^{\mathrm{T}}}$ in $d$
9:    $\omega_{y^{\mathrm{S}}, y^{\mathrm{T}}} \leftarrow 1$
10:    $d_{y^{\mathrm{S}}, t} \leftarrow 0$ for $t = 1, 2, ..., k_{\mathrm{T}}$
11:    $d_{s, y^{\mathrm{T}}} \leftarrow 0$ for $s = 1, 2, ..., k_{\mathrm{S}}$
12:    $T \leftarrow T \cup \{y^{\mathrm{S}}\}$
13: **end while**
14: **return** $\omega$
---

maximum value in the rows and columns corresponding to unmatched label pairs in the matrix to determine the one-to-one mappings. FLM is also computed prior to the training of $f_{\mathrm{in}}(\cdot; \theta)$.

### D.3 Iterative Label Mapping (ILM)

---
**Algorithm 6** Iterative Label Mapping for VR
---
1: **Input:** Pretrained label space $\mathcal{Y}^{\mathrm{S}}$ with $k_{\mathrm{S}}$ labels, downstream label space $\mathcal{Y}^{\mathrm{T}}$ with $k_{\mathrm{T}}$ labels, downstream training set $\{(x_i^{\mathrm{T}}, y_i^{\mathrm{T}})\}_{i=1}^n$, given pretrained model $f_{\mathrm{pre}}(\cdot)$, total iteration number $E$, learning rate $a$
2: **Output:** One-to-one LM matrix $\omega \in \{0,1\}^{k_{\mathrm{S}} \times k_{\mathrm{T}}}$
3: Initialize $\omega \leftarrow \{0\}^{k_{\mathrm{S}} \times k_{\mathrm{T}}}$, temp set $T \leftarrow \{\}$ to store matched pretrained labels, initialize $f_{\mathrm{in}}(\cdot; \theta)$ ($\theta \leftarrow \mathbf{0}$)
4: **for** $e = 1...E$ **do**
5:    # Computing frequency distribution matrix $d$
6:    Use Algorithm 4 to obtain $d$
7:    # Computing output mapping $\omega$
8:    **while** size of $T$ is not $k_{\mathrm{T}}$ **do**
9:       Find the maximum $d_{y^{\mathrm{S}}, y^{\mathrm{T}}}$ in $d$
10:       $\omega_{y^{\mathrm{S}}, y^{\mathrm{T}}} \leftarrow 1$
11:       $d_{y^{\mathrm{S}}, t} \leftarrow 0$ for $t = 1, 2, ..., k_{\mathrm{T}}$
12:       $d_{s, y^{\mathrm{T}}} \leftarrow 0$ for $s = 1, 2, ..., k_{\mathrm{S}}$
13:       $T \leftarrow T \cup \{y^{\mathrm{S}}\}$
14:    **end while**
15:    # Training $f_{\mathrm{in}}(\cdot; \theta)$
16:    $\theta \leftarrow \theta - a \cdot \nabla_\theta \sum_{i=1}^n \ell(y_i^{\mathrm{T}}, f_{\mathrm{out}}^\omega(f_{\mathrm{pre}}(f_{\mathrm{in}}(x_i^{\mathrm{T}}; \theta))))$
17: **end for**
18: **return** $\omega$
---

As an enhanced version of FLM, *iterative label mapping* (ILM) employs interleaved updates with $\theta$ at each epoch, as outlined in Algorithm 6. Such interleaved updates take into consideration the variations in the output space induced by updates to the input VR during the training process, thereby ensuring that the output LM will be matched with the updated VR.

### D.4 Bayesian-guided Label Mapping (BLM)

The detailed procedure of *Bayesian-guided label mapping* (BLM) proposed in this paper is shown in Algorithm 7. Compared to ILM, BLM replaces the previous one-to-one mapping $\omega \in \{0,1\}^{k_{\mathrm{S}} \times k_{\mathrm{T}}}$

with probabilistic LM $\omega \in [0,1]^{k_{\mathrm{S}} \times k_{\mathrm{T}}}$, both satisfying $\sum_{j=1}^{k_{\mathrm{S}}} \omega_{j,\cdot} = 1$ as stated in Section 3. Meanwhile, the process of matrix computation for BLM is based on the Bayes' theorem (detailed in Section 4) to reflect the complex relationship among label spaces, rather than determining the optimal match through the oversimplified greedy algorithm.

---

**Algorithm 7** Bayesian-guided Label Mapping for VR

---

1: **Input:** Pretrained label space $\mathcal{Y}^{\mathrm{S}}$ with $k_{\mathrm{S}}$ labels, downstream label space $\mathcal{Y}^{\mathrm{T}}$ with $k_{\mathrm{T}}$ labels, downstream training set $\{(x_i^{\mathrm{T}}, y_i^{\mathrm{T}})\}_{i=1}^n$, given pretrained model $f_{\mathrm{pre}}(\cdot)$, total iteration number $E$, learning rate $a$, Laplace smoothing $\lambda$
2: **Output:** Probabilistic LM $\omega \in [0,1]^{k_{\mathrm{S}} \times k_{\mathrm{T}}}$
3: Initialize $\omega \leftarrow \{0\}^{k_{\mathrm{S}} \times k_{\mathrm{T}}}$, initialize $f_{\mathrm{in}}(\cdot; \theta)$ ($\theta \leftarrow \mathbf{0}$), temp matrix $P = [P_1, ..., P_{k_{\mathrm{S}}}]^\top \in \mathbb{R}^{k_{\mathrm{S}}}$
4: **for** $e = 1...E$ **do**
5:     # Computing frequency distribution matrix $d$
6:     Use Algorithm 4 to obtain $d$
7:     # Computing output mapping $\omega$
8:     $P_{y^{\mathrm{S}}} \leftarrow \sum_{t=1}^{k_{\mathrm{T}}} d_{y^{\mathrm{S}}, t} + \lambda$ for $y^{\mathrm{S}} = 1...k_{\mathrm{S}}$
9:     $\omega_{y^{\mathrm{S}}, y^{\mathrm{T}}} \leftarrow d_{y^{\mathrm{S}}, y^{\mathrm{T}}} / P_{y^{\mathrm{S}}}$ for $y^{\mathrm{S}} = 1...k_{\mathrm{S}}, y^{\mathrm{T}} = 1...k_{\mathrm{T}}$
10:     # Column normalization of $\omega$
11:     $\omega_{y^{\mathrm{S}}, y^{\mathrm{T}}} \leftarrow \omega_{y^{\mathrm{S}}, y^{\mathrm{T}}} / \sum_{s=1}^{k_{\mathrm{S}}} \omega_{s, y^{\mathrm{T}}}$ for $y^{\mathrm{S}} = 1...k_{\mathrm{S}}, y^{\mathrm{T}} = 1...k_{\mathrm{T}}$
12:     # Training $f_{\mathrm{in}}(\cdot; \theta)$
13:     $\theta \leftarrow \theta - a \cdot \nabla_\theta \sum_{i=1}^n \ell(y_i^{\mathrm{T}}, f_{\mathrm{out}}^\omega(f_{\mathrm{pre}}(f_{\mathrm{in}}(x_i^{\mathrm{T}}; \theta))))$
14: **end for**
15: **return** $\omega$

---

## D.5 Improved Bayesian-guided Label Mapping (BLM+)

---

**Algorithm 8** Computing Probability Aggregation Matrix by Top-$K$ Predicted Probabilities

---

1: **Input:** Downstream training set $\{(x_i^{\mathrm{T}}, y_i^{\mathrm{T}})\}_{i=1}^n$, given input VR $f_{\mathrm{in}}(\cdot; \theta)$ and pretrained model $f_{\mathrm{pre}}(\cdot)$ with the $j$th dimension being $f_{\mathrm{pre}}(\cdot)_j$, Laplace smoothing $\lambda$, top-$K$ value $k$
2: **Output:** Probability aggregation matrix $d' \in \mathbb{R}^{k_{\mathrm{S}} \times k_{\mathrm{T}}}$
3: Initialize $d' \leftarrow \{0\}^{k_{\mathrm{S}} \times k_{\mathrm{T}}}$, temp matrix $Q = [Q_1, ..., Q_k]^\top \in \mathbb{R}^k$, $K = [K_1, ..., K_k]^\top \in \mathbb{N}^{+k}$
4: # Computing aggregation distribution matrix $d'$
5: **for** $i = 1...n$ **do**
6:     $Q \leftarrow \mathrm{TopK}_j(\mathrm{softmax}(f_{\mathrm{pre}}(f_{\mathrm{in}}(x_i^{\mathrm{T}}; \theta))_j), k)$  # top-$K$
7:     $K \leftarrow \mathrm{TopKIndices}_j(f_{\mathrm{pre}}(f_{\mathrm{in}}(x_i^{\mathrm{T}}; \theta))_j, k)$
8:     $d'_{K_s, y_i^{\mathrm{T}}} \leftarrow d'_{K_s, y_i^{\mathrm{T}}} + Q_s$ for $s = 1...k$  # Probability Aggregation
9: **end for**
10: **return** $d'$

---

As mentioned in Section 4, BLM+ extends BLM by incorporating the aggregation of top-$K$ predicted probabilities, shown in Algorithm 9.

This divergence manifests in the computation process of the joint distribution matrix between predicted pretrained labels and ground-truth downstream labels. Previous methods (i.e., RLM, ILM, BLM) computed a non-negative integer matrix $d \in \mathbb{Z}^{k_{\mathrm{S}} \times k_{\mathrm{T}}}$ based on the frequency of occurrence of samples (Algorithm 4).

In BLM+, the calculation entails replacing the deterministic frequencies with predicted probabilities from the top $K$ pretrained labels to estimate the joint distribution matrix $d \in \mathbb{R}^{k_{\mathrm{S}} \times k_{\mathrm{T}}}$, as is shown in Algorithm 8. In the procedure, the probability aggregation substitutes the binary frequency distribution $\{0, 1\}$ with a probability distribution within the range of $[0, 1]$, while the top-$K$ technique serves to retain the most probable $k$ predicted labels rather than selecting only one (i.e., BLM) or all labels (denoted as '-Top-K' in ablation studies in Section 5).

---

**Algorithm 9** Improved Bayesian-guided Label Mapping for VR

---

1: **Input:** Pretrained label space $\mathcal{Y}^{\mathrm{S}}$ with $k_{\mathrm{S}}$ labels, downstream label space $\mathcal{Y}^{\mathrm{T}}$ with $k_{\mathrm{T}}$ labels, downstream training set $\{(x_i^{\mathrm{T}}, y_i^{\mathrm{T}})\}_{i=1}^n$, given pretrained model $f_{\mathrm{pre}}(\cdot)$ with the $j$th dimension being $f_{\mathrm{pre}}(\cdot)_j$, total iteration number $E$, learning rate $a$, Laplace smoothing $\lambda$, top-$K$ value $k$

2: **Output:** Probabilistic LM $\omega \in [0, 1]^{k_{\mathrm{S}} \times k_{\mathrm{T}}}$

3: Initialize $\omega \leftarrow \{0\}^{k_{\mathrm{S}} \times k_{\mathrm{T}}}$, initialize $f_{\mathrm{in}}(\cdot; \theta)$ $(\theta \leftarrow \mathbf{0})$, temp matrix $P = [P_1, ..., P_{k_{\mathrm{S}}}]^\top \in \mathbb{R}^{k_{\mathrm{S}}}$

4: **for** $e = 1...E$ **do**

5:      # Computing probability aggregation matrix $d'$

6:      Use Algorithm 8 to obtain $d'$

7:      # Computing output mapping $\omega$

8:      $P_{y^{\mathrm{S}}} \leftarrow \sum_{t=1}^{k_{\mathrm{T}}} d_{y^{\mathrm{S}}, t} + \lambda$ for $y^{\mathrm{S}} = 1...k_{\mathrm{S}}$

9:      $\omega_{y^{\mathrm{S}}, y^{\mathrm{T}}} \leftarrow d_{y^{\mathrm{S}}, y^{\mathrm{T}}} / P_{y^{\mathrm{S}}}$ for $y^{\mathrm{S}} = 1...k_{\mathrm{S}}, y^{\mathrm{T}} = 1...k_{\mathrm{T}}$

10:      # Column normalization of $\omega$

11:      $\omega_{y^{\mathrm{S}}, y^{\mathrm{T}}} \leftarrow \omega_{y^{\mathrm{S}}, y^{\mathrm{T}}} / \sum_{s=1}^{k_{\mathrm{S}}} \omega_{s, y^{\mathrm{T}}}$ for $y^{\mathrm{S}} = 1...k_{\mathrm{S}}, y^{\mathrm{T}} = 1...k_{\mathrm{T}}$

12:      # Training $f_{\mathrm{in}}(\cdot; \theta)$

13:      $\theta \leftarrow \theta - a \cdot \nabla_\theta \sum_{i=1}^n \ell(y_i^{\mathrm{T}}, f_{\mathrm{out}}^\omega(f_{\mathrm{pre}}(f_{\mathrm{in}}(x_i^{\mathrm{T}}; \theta))))$

14: **end for**

15: **return** $\omega$

---

## D.6 A Quick Version of ILM, BLM, and BLM+

The baseline method FLM calculates the mapping $\omega$ once and keeps it fixed, while ILM and our methods update $\omega$ at each step. However, updating $\omega$ does not require running the model twice to obtain current predictions for each epoch. Instead, predictions from the most recent epoch can be reused. Therefore, only in the first epoch is it necessary to run the pretrained model an additional time to initialize the weights of LM, which is the same as FLM. In subsequent epochs, these methods do not require any extra runs. More details can be found in the quick version of our released code.

# E Detailed Theoretical Analysis

## E.1 Justification and Analysis

In this section, we investigate why probabilistic LM should be favored over deterministic one-to-one mapping. This analysis assumes the existence of true correspondences between labels in the pretrained and downstream domains. We establish that, under certain conditions, probabilistic LM (Definition. E.1) outperforms deterministic LM (Definition E.2) in estimating the distribution of true label correspondences, quantified by the expected accuracy of the LM function (Eq. (10)).

This analysis focuses on the comparisons of LM. Given that the pretrained model $f_{\mathrm{pre}}$, input $x$, and input transformations $f_{\mathrm{in}}$ are the same across different LM methods, we will omit these notations below unless explicitly needed. We begin by introducing key definitions.

**Definition E.1** (*probabilistic label mapping* (PLM)). *Let $\mathcal{F}_{\mathrm{plm}} \subset \mathcal{F}_{\mathrm{lm}}$ be a set of mapping functions such that for all $f_{\mathrm{plm}} \in \mathcal{F}_{\mathrm{plm}}$, we have*

$$p(f_{\mathrm{plm}}(y^{\mathrm{S}}) = y^{\mathrm{T}} | y^{\mathrm{S}}) = \omega_{y^{\mathrm{S}}, y^{\mathrm{T}}}, \ s.t. \sum_{y^{\mathrm{S}} \in \mathcal{Y}^{\mathrm{S}}} \omega_{y^{\mathrm{S}}, y^{\mathrm{T}}} = 1, \forall y^{\mathrm{T}} \in \mathcal{Y}^{\mathrm{T}}. \tag{11}$$

*Here, $p(f_{\mathrm{plm}}(y^{\mathrm{S}}) = y^{\mathrm{T}} | y^{\mathrm{S}})$ is the conditional probability that a pretrained label $y^{\mathrm{S}}$ is mapped to a downstream label $y^{\mathrm{T}}$.*

**Definition E.2** (*deterministic label mapping* (DLM)). *Let $\mathcal{F}_{\mathrm{dlm}} \subset \mathcal{F}_{\mathrm{lm}}$ be a set of mapping functions, defined by $f_{\mathrm{dlm}}(y^{\mathrm{S}}) = g(y^{\mathrm{S}})$ for all $y^{\mathrm{S}} \in \mathcal{Y}^{\mathrm{S}}$, where $g(y^{\mathrm{S}})$ specifies a deterministic rule, either $g(y^{\mathrm{S}}) = y^{\mathrm{S}}$ for identity mapping; or $g(y^{\mathrm{S}}) = 1 - y^{\mathrm{S}}$ for flip mapping, respectively. Then, deterministic label mapping is defined as: $\forall f_{\mathrm{dlm}} \in \mathcal{F}_{\mathrm{dlm}}$,*

$$p(f_{\mathrm{dlm}}(y^{\mathrm{S}}) = y^{\mathrm{T}} | y^{\mathrm{S}}) = \delta_{y^{\mathrm{S}}, g(y^{\mathrm{S}})}, \ with \ \delta_{y^{\mathrm{S}}, g(y^{\mathrm{S}})} = \begin{cases} 1 & if \ g(y^{\mathrm{S}}) = y^{\mathrm{T}} \\ 0 & otherwise \end{cases}, \tag{12}$$

*where $\delta$ is the Kronecker delta function, ensuring $y^{\mathrm{T}}$ is uniquely mapped from a pretrained label $y^{\mathrm{S}}$.*

Then, we demonstrate the conditions where $\text{Acc}(f_{\text{plm}}) \geq \text{Acc}(f_{\text{dlm}})$. Since DLM is defined by $g$, following either identity mapping or flip mapping exclusively, each case will be discussed separately.

**Lemma E.3.** *Given a collection of paired labels $\{(y^{\text{S}}, y^{\text{T}})\}_{i=1}^n$. If the aggregate conditional probabilities $p(y^{\text{S}} = 1 | y^{\text{T}} = 0) \geq p(y^{\text{S}} = 0 | y^{\text{T}} = 0)$ and $p(y^{\text{S}} = 0 | y^{\text{T}} = 1) \geq p(y^{\text{S}} = 1 | y^{\text{T}} = 1)$ hold true, and considering $f_{\text{dlm}}$ is defined by identity mapping as outlined in Definition E.2, then it follows that $\text{Acc}(f_{\text{plm}}) \geq \text{Acc}(f_{\text{dlm}})$.*

Lemma E.3 (proof in Appendix E.2) implies that PLM achieves at least as high expected accuracy as DLM defined by identity mapping, under the following conditions: for downstream samples with $y^{\text{T}} = 0$, the inequality is satisfied when they are more likely to correspond to pretrained samples with $y^{\text{S}} = 1$ than those with $y^{\text{S}} = 0$; for downstream samples with $y^{\text{T}} = 1$, the inequality is satisfied when the corresponding pretrained samples are more likely to have $y^{\text{S}} = 0$ than $y^{\text{S}} = 1$.

**Uncertainty in Label Inter-Dependencies**. Essentially, the conditions above reflect potential complex patterns of label correspondence that arise when inter-dependencies between the labels exist across domains. While this "label mismatch" problem has been discussed in binary settings, it can be generalized to multi-class settings without loss of generality. Unlike DLM, which merely relies on a static mapping rule and hence may fail when true label correspondence conflicts with this predefined rule, PLM captures the conditional probabilities of $y^{\text{T}}$ given $y^{\text{S}}$. By harnessing the inherent uncertainty encoded in the probabilistic form of $\omega$, PLM is expected to achieve more robust label mapping predictions.

Next, we compare PLM with DLM using the flip mapping rule.

**Lemma E.4.** *Given a collection of paired labels $\{(y^{\text{S}}, y^{\text{T}})\}_{i=1}^n$. If the aggregate conditional probabilities $p(y^{\text{S}} = 0 | y^{\text{T}} = 0) \leq p(y^{\text{S}} = 1 | y^{\text{T}} = 0)$ and $p(y^{\text{S}} = 0 | y^{\text{T}} = 1) \leq p(y^{\text{S}} = 1 | y^{\text{T}} = 1)$, and $f_{\text{dlm}}$ is defined by flip mapping as outlined in Definition E.2, then $\text{Acc}(f_{\text{plm}}) \geq \text{Acc}(f_{\text{dlm}})$.*

Lemma E.4 (proof in Appendix E.2) establishes another sufficient condition under which PLM could achieve an expected accuracy at least as high as DLM defined by flip mapping. The condition applies to all downstream samples, regardless of their labels (both $y^{\text{T}} = 0$ or $y^{\text{T}} = 1$), stating that it is more likely that their corresponding pretrained label being $y^{\text{S}} = 1$ rather than $y^{\text{S}} = 0$.

**Bias in Label Correspondences**. The bias in Label correspondence refers to a phenomenon where a disproportionate number of downstream samples correspond to pretrained samples with a specific label. For example, consider a medical diagnosis task where both pretrained and downstream data come from populations with low disease prevalence, the label correspondences may exhibit this bias [55]. While this bias may be overlooked by DLM, it could be captured and even exploited by PLM, which flexibly adjusts the weighting schemes, e.g., assigning higher value to $\omega_{1,0}$ than $\omega_{0,0}$ for samples where $y^{\text{T}} = 0$, and to $\omega_{1,1}$ over $\omega_{0,1}$ for samples where $y^{\text{T}} = 1$.

**Corollary E.5.** *Let $f_{\text{plm}}$ and $f_{\text{dlm}}$ denote the label mapping functions defined in Definition E.1 and Definition E.2, respectively. Given pretrained and downstream label spaces $\mathcal{Y}^{\text{S}} = \{0, 1\}$ and $\mathcal{Y}^{\text{T}} = \{0, 1\}$, if for any joint distribution over $\mathcal{Y}^{\text{S}} \times \mathcal{Y}^{\text{T}}$,*

$$\exists\, a \in \{0, 1\} \; s.t. \; p(y^{\text{S}} = a | y^{\text{T}} = \bar{a}) \geq p(y^{\text{S}} = \bar{a} | y^{\text{T}} = \bar{a}), \tag{13}$$

*where $\bar{a}$ is the opposite label of $a$, then we have $\text{Acc}(f_{\text{plm}}) \geq \text{Acc}(f_{\text{dlm}})$.*

*Remark* E.6. Corollary E.5 implies a theoretical foundation for preferring PLM over DLM in scenarios where the label mapping relationship between two domains is uncertain, biased and potentially deviates from a deterministic one-to-one mapping assumption. This finding holds importance in label mappings for VR, as the label spaces may encompass multi-class settings. Furthermore, in VR settings, the pretrained labels derived from $f_{\text{pre}}$ predictions, are subject to increased uncertainties and biases influenced by the quality and distribution of the pretrained model and dataset[6].

## E.2 Completed Proof of Lemma E.3 and Lemma E.4

**Lemma E.7** (*cf.* Lemma E.3). *Given a collection of paired labels $\{(y^{\text{S}}, y^{\text{T}})\}_{i=1}^n$. If the aggregate conditional probabilities $p(y^{\text{S}} = 1 | y^{\text{T}} = 0) \geq p(y^{\text{S}} = 0 | y^{\text{T}} = 0)$ and $p(y^{\text{S}} = 0 | y^{\text{T}} = 1) \geq p(y^{\text{S}} =$*

---

[6]For the analysis purpose, in this section we simplify the setting and operate with ground-truth $\mathcal{Y}^{\text{S}}$. In practice, VR does not have access to true $y^{\text{S}}$ but must rely on the predicted $y^{\text{S}}$ from the well-trained $f_{\text{pre}}$ instead.

$1|y^{\mathrm{T}} = 1)$ *hold true, and $f_{\mathrm{dlm}}$ is defined by identity mapping as outlined in Definition E.2, then* $\mathrm{Acc}(f_{\mathrm{plm}}) \geq \mathrm{Acc}(f_{\mathrm{dlm}})$.

*Proof.* Expand Eq. (10) by taking all possibilities of $y^{\mathrm{T}}$, we have:

$$
\begin{aligned}
\mathrm{Acc}(f_{\mathrm{lm}}) &= \mathbb{E}_{y^{\mathrm{T}} \in \mathcal{Y}^{\mathrm{T}}} \left[ \sum_{y^{\mathrm{S}} \in \mathcal{Y}^{\mathrm{S}}} p(y^{\mathrm{S}}) \cdot p\left(f_{\mathrm{lm}}(y^{\mathrm{S}}) = y^{\mathrm{T}} | y^{\mathrm{S}}\right) \right] \\
&= \sum_{y^{\mathrm{T}} \in \mathcal{Y}^{\mathrm{T}}} p(y^{\mathrm{T}}) \left[ \sum_{y^{\mathrm{S}} \in \mathcal{Y}^{\mathrm{S}}} p(y^{\mathrm{S}} | y^{\mathrm{T}}) \cdot p\left(f_{\mathrm{lm}}(y^{\mathrm{S}}) = y^{\mathrm{T}} | y^{\mathrm{S}}, y^{\mathrm{T}}\right) \right] \\
&= \sum_{y^{\mathrm{T}} \in \mathcal{Y}^{\mathrm{T}}} p(y^{\mathrm{T}}) \left[ \sum_{y^{\mathrm{S}} \in \mathcal{Y}^{\mathrm{S}}} p(y^{\mathrm{S}} | y^{\mathrm{T}}) \cdot p\left(f_{\mathrm{lm}}(y^{\mathrm{S}}) = y^{\mathrm{T}} | y^{\mathrm{S}}\right) \right].
\end{aligned}
\tag{14}
$$

Note that the conditional independence holds since the output of $f_{\mathrm{lm}}$ relies solely on the input $y^{\mathrm{S}}$. For DLM defined by identity mapping, $p(f_{\mathrm{dlm}}(y^{\mathrm{S}}) = y^{\mathrm{T}} | y^{\mathrm{S}}) = 1$ if $y^{\mathrm{S}} = y^{\mathrm{T}}$, and 0 otherwise. Taking into account all the samples, the expected accuracy $\mathrm{Acc}(f_{\mathrm{dlm}})$ can then be expressed by

$$
\begin{aligned}
\mathrm{Acc}(f_{\mathrm{dlm}}) &= \sum_{y^{\mathrm{T}} \in \mathcal{Y}^{\mathrm{T}}} p(y^{\mathrm{T}}) \, p(y^{\mathrm{S}} = y^{\mathrm{T}} | y^{\mathrm{T}}) \\
&= p(y^{\mathrm{T}} = 0) p(y^{\mathrm{S}} = 0 | y^{\mathrm{T}} = 0) + p(y^{\mathrm{T}} = 1) p(y^{\mathrm{S}} = 1 | y^{\mathrm{T}} = 1).
\end{aligned}
\tag{15}
$$

As for PLM, the expected accuracy can be rewritten as

$$
\begin{aligned}
\mathrm{Acc}(f_{\mathrm{plm}}) &= \sum_{y^{\mathrm{T}} \in \mathcal{Y}^{\mathrm{T}}} p(y^{\mathrm{T}}) \sum_{y^{\mathrm{S}} \in \mathcal{Y}^{\mathrm{S}}} \omega_{y^{\mathrm{S}}, y^{\mathrm{T}}} \cdot p(y^{\mathrm{S}} | y^{\mathrm{T}}) \\
&= p(y^{\mathrm{T}} = 0) \left[ \omega_{0,0} \cdot p(y^{\mathrm{S}} = 0 | y^{\mathrm{T}} = 0) + \omega_{1,0} \cdot p(y^{\mathrm{S}} = 1 | y^{\mathrm{T}} = 0) \right] \\
&\quad + p(y^{\mathrm{T}} = 1) \left[ \omega_{0,1} \cdot p(y^{\mathrm{S}} = 0 | y^{\mathrm{T}} = 1) + \omega_{1,1} \cdot p(y^{\mathrm{S}} = 1 | y^{\mathrm{T}} = 1) \right],
\end{aligned}
\tag{16}
$$

where $\omega_{0,0}$ stands for $\omega_{y^{\mathrm{S}} = 0, y^{\mathrm{T}} = 0}$, and similarly for the remaining $\omega_{0,1}, \omega_{1,0}, \omega_{1,1}$.

To evaluate the expected accuracy of $f_{\mathrm{plm}}$ and $f_{\mathrm{dlm}}$, we look into the comparison separately for each $y^{\mathrm{T}}$. Specifically, for the samples with $y^{\mathrm{T}} = 0$, we aim to show that

$$
\begin{aligned}
p(y^{\mathrm{T}} = 0) &\left[ \omega_{0,0} \cdot p(y^{\mathrm{S}} = 0 | y^{\mathrm{T}} = 0) + \omega_{1,0} \cdot p(y^{\mathrm{S}} = 1 | y^{\mathrm{T}} = 0) \right] \\
&\geq p(y^{\mathrm{T}} = 0) p(y^{\mathrm{S}} = 0 | y^{\mathrm{T}} = 0).
\end{aligned}
\tag{17}
$$

Given the constraints $\omega_{0,0} + \omega_{1,0} = 1$ and $p(y^{\mathrm{S}} = 0 | y^{\mathrm{T}} = 0) + p(y^{\mathrm{S}} = 1 | y^{\mathrm{T}} = 0) = 1$, the LHS of Eq. (17) becomes

$$
\begin{aligned}
&p(y^{\mathrm{T}} = 0) \left[ \omega_{0,0} \cdot p(y^{\mathrm{S}} = 0 | y^{\mathrm{T}} = 0) + \omega_{1,0} \cdot p(y^{\mathrm{S}} = 1 | y^{\mathrm{T}} = 0) \right] \\
&= p(y^{\mathrm{T}} = 0) \left[ (\omega_{0,0} \cdot p(y^{\mathrm{S}} = 0 | y^{\mathrm{T}} = 0) + \omega_{1,0}(1 - p(y^{\mathrm{S}} = 0 | y^{\mathrm{T}} = 0))) \right] \\
&= p(y^{\mathrm{T}} = 0) \left[ (\omega_{0,0} - \omega_{1,0}) \cdot p(y^{\mathrm{S}} = 0 | y^{\mathrm{T}} = 0) + \omega_{1,0} \right].
\end{aligned}
\tag{18}
$$

The inequality we need to show is then simplified to $p(y^{\mathrm{T}} = 0)[(\omega_{0,0} - \omega_{1,0}) \cdot p(y^{\mathrm{S}} = 0 | y^{\mathrm{T}} = 0) + \omega_{1,0}] \geq p(y^{\mathrm{T}} = 0) p(y^{\mathrm{S}} = 0 | y^{\mathrm{T}} = 0)$. This inequality holds if $p(y^{\mathrm{S}} = 0 | y^{\mathrm{T}} = 0) \leq p(y^{\mathrm{S}} = 1 | y^{\mathrm{T}} = 0)$.

Similarly, for samples with $y^{\mathrm{T}} = 1$, the inequality of interest is

$$
\begin{aligned}
p(y^{\mathrm{T}} = 1) &\left[ \omega_{1,0} \cdot p(y^{\mathrm{S}} = 1 | y^{\mathrm{T}} = 0) + \omega_{1,1} \cdot p(y^{\mathrm{S}} = 1 | y^{\mathrm{T}} = 1) \right] \\
&\geq p(y^{\mathrm{T}} = 1) p(y^{\mathrm{S}} = 1 | y^{\mathrm{T}} = 1).
\end{aligned}
\tag{19}
$$

This holds if $p(y^{\mathrm{S}} = 1 | y^{\mathrm{T}} = 1) \leq p(y^{\mathrm{S}} = 0 | y^{\mathrm{T}} = 1)$.

Both conditions can be satisfied without conflict. Thus, we can confirm Lemma E.3 by evaluating these conditions jointly. $\qquad \square$

**Lemma E.8** (*cf.* Lemma E.4). *Given a collection of paired labels $\{(y^{\mathrm{S}}, y^{\mathrm{T}})\}_{i=1}^n$. If the aggregate conditional probabilities $p(y^{\mathrm{S}} = 0|y^{\mathrm{T}} = 0) \leq p(y^{\mathrm{S}} = 1|y^{\mathrm{T}} = 0)$ and $p(y^{\mathrm{S}} = 0|y^{\mathrm{T}} = 1) \leq p(y^{\mathrm{S}} = 1|y^{\mathrm{T}} = 1)$, and $f_{\mathrm{dlm}}$ is defined by flip mapping as outlined in Definition E.2, then $\mathrm{Acc}(f_{\mathrm{plm}}) \geq \mathrm{Acc}(f_{\mathrm{dlm}})$.*

*Proof.* When defined deterministically by flip mapping, DLM can be equivalently expressed as $p(f_{\mathrm{dlm}}(y^{\mathrm{S}}) = y^{\mathrm{T}}|y^{\mathrm{S}}) = 1$ if $y^{\mathrm{S}} \neq y^{\mathrm{T}}$, and $0$ otherwise. This allows the expected accuracy of DLM to be expanded as:

$$
\begin{aligned}
\mathrm{Acc}(f_{\mathrm{dlm}}) &= \sum_{y^{\mathrm{T}} \in \mathcal{Y}^{\mathrm{T}}} p(y^{\mathrm{T}})\, p(y^{\mathrm{S}} \neq y^{\mathrm{T}}|y^{\mathrm{T}}) \\
&= \sum_{y^{\mathrm{T}} \in \mathcal{Y}^{\mathrm{T}}} p(y^{\mathrm{T}}) \left(1 - p(y^{\mathrm{S}} = y^{\mathrm{T}}|y^{\mathrm{T}})\right) \\
&= p(y^{\mathrm{T}} = 0) \left(1 - p(y^{\mathrm{S}} = 1|y^{\mathrm{T}} = 0)\right) + p(y^{\mathrm{T}} = 1) \left(1 - p(y^{\mathrm{S}} = 0|y^{\mathrm{T}} = 1)\right).
\end{aligned} \tag{20}
$$

Meanwhile, the expected accuracy of PLM remains consistent as in Eq. (16). Again, to show that $\mathrm{Acc}(f_{\mathrm{plm}}) \geq \mathrm{Acc}(f_{\mathrm{dlm}})$ holds, we compare the expected accuracy with respect to different $y^{\mathrm{T}}$ samples separately.

For samples with $y^{\mathrm{T}} = 0$, we need to show

$$
\begin{aligned}
p(y^{\mathrm{T}} = 0) \left[\omega_{0,0} \cdot p(y^{\mathrm{S}} = 0|y^{\mathrm{T}} = 0) + \omega_{1,0} \cdot p(y^{\mathrm{S}} = 1|y^{\mathrm{T}} = 0)\right] \\
\geq p(y^{\mathrm{T}} = 0)p(y^{\mathrm{T}} = 0) \left(1 - p(y^{\mathrm{S}} = 1|y^{\mathrm{T}} = 0)\right).
\end{aligned} \tag{21}
$$

Given the constraints that $\omega_{0,0} + \omega_{1,0} = 1$ and $p(y^{\mathrm{S}} = 1|y^{\mathrm{T}} = 0) = 1 - p(y^{\mathrm{S}} = 0|y^{\mathrm{T}} = 0)$, the LHS of Eq. (21) can be expressed by

$$
\begin{aligned}
p(y^{\mathrm{T}} = 0) \left[\omega_{0,0} \cdot p(y^{\mathrm{S}} = 0|y^{\mathrm{T}} = 0) + \omega_{1,0} \cdot p(y^{\mathrm{S}} = 1|y^{\mathrm{T}} = 0)\right] \\
= p(y^{\mathrm{T}} = 0) \left[(\omega_{0,0} - \omega_{1,0}) \cdot p(y^{\mathrm{S}} = 0|y^{\mathrm{T}} = 0) + \omega_{1,0}\right].
\end{aligned} \tag{22}
$$

We rearrange the terms:

$$
\begin{aligned}
p(y^{\mathrm{T}} = 0) \left[(\omega_{0,0} - \omega_{1,0}) \cdot p(y^{\mathrm{S}} = 0|y^{\mathrm{T}} = 0) + \omega_{1,0}\right] &\geq p(y^{\mathrm{T}} = 0) \left(1 - p(y^{\mathrm{S}} = 0|y^{\mathrm{T}} = 0)\right) \\
(\omega_{0,0} - \omega_{1,0}) \cdot p(y^{\mathrm{S}} = 0|y^{\mathrm{T}} = 0) + \omega_{1,0} &\geq 1 - p(y^{\mathrm{S}} = 0|y^{\mathrm{T}} = 0) \\
\frac{1 - p(y^{\mathrm{S}} = 0|y^{\mathrm{T}} = 0) - \omega_{1,0} + \omega_{1,0} \cdot p(y^{\mathrm{S}} = 0|y^{\mathrm{T}} = 0)}{p(y^{\mathrm{S}} = 0|y^{\mathrm{T}} = 0)} &\leq \omega_{0,0} \\
\frac{1 - p(y^{\mathrm{S}} = 0|y^{\mathrm{T}} = 0) - \omega_{1,0} \cdot (1 - p(y^{\mathrm{S}} = 0|y^{\mathrm{T}} = 0))}{p(y^{\mathrm{S}} = 0|y^{\mathrm{T}} = 0)} &\leq \omega_{0,0} \\
\frac{(1 - \omega_{1,0}) \cdot (1 - p(y^{\mathrm{S}} = 0|y^{\mathrm{T}} = 0))}{p(y^{\mathrm{S}} = 0|y^{\mathrm{T}} = 0)} &\leq \omega_{0,0}.
\end{aligned} \tag{23}
$$

It is then concluded that Eq. (23) holds if $p(y^{\mathrm{S}} = 0|y^{\mathrm{T}} = 0) \leq p(y^{\mathrm{S}} = 1|y^{\mathrm{T}} = 0)$.

As with $y^{\mathrm{T}} = 1$ samples, a similar derivation is performed to satisfy the inequality

$$
\begin{aligned}
p(y^{\mathrm{T}} = 1) \left[\omega_{0,1} \cdot p(y^{\mathrm{S}} = 0|y^{\mathrm{T}} = 1) + \omega_{1,1} \cdot p(y^{\mathrm{S}} = 1|y^{\mathrm{T}} = 1)\right] \\
\geq p(y^{\mathrm{T}} = 0)p(y^{\mathrm{T}} = 0) \left(1 - p(y^{\mathrm{S}} = 1|y^{\mathrm{T}} = 0)\right).
\end{aligned} \tag{24}
$$

Resembling $y^{\mathrm{T}} = 0$ samples, the derivation yields the condition $p(y^{\mathrm{S}} = 0|y^{\mathrm{T}} = 1) \leq p(y^{\mathrm{S}} = 1|y^{\mathrm{T}} = 1)$.

Notably, the condition $p(y^{\mathrm{S}} = 0|y^{\mathrm{T}} = 0) \leq p(y^{\mathrm{S}} = 1|y^{\mathrm{T}} = 0)$ does not conflict with $p(y^{\mathrm{S}} = 0|y^{\mathrm{T}} = 1) \leq p(y^{\mathrm{S}} = 1|y^{\mathrm{T}} = 1)$, and both conditions can be jointly satisfied. $\qquad\square$

# F  Training Details

## F.1  Dataset Information

Additional dataset information is presented in Table 4. For a fair comparison, we adhere to the data partitioning scheme employed by ILM [4] through all datasets. The batch size for Oxfordpets and DTD is set to be 64 while 256 for the remaining datasets.

Table 4: Detailed dataset information

| Dataset | Original Image Size | Training Set Size | Testing Set Size | Number of Classes |
|---|---|---|---|---|
| Flowers102 | $128 \times 128$ | 4,093 | 2,463 | 102 |
| DTD | $128 \times 128$ | 2,820 | 1,692 | 47 |
| UCF101 | $128 \times 128$ | 7,639 | 3,783 | 101 |
| Food101 | $128 \times 128$ | 50,500 | 30,300 | 101 |
| GTSRB | $32 \times 32$ | 39,209 | 12,630 | 43 |
| EuroSAT | $128 \times 128$ | 13,500 | 8,100 | 10 |
| OxfordPets | $128 \times 128$ | 2,944 | 3,669 | 37 |
| StanfordCars | $128 \times 128$ | 6,509 | 8,041 | 196 |
| SUN397 | $128 \times 128$ | 15,888 | 19,850 | 397 |
| CIFAR10 | $32 \times 32$ | 50,000 | 10,000 | 10 |
| CIFAR100 | $32 \times 32$ | 50,000 | 10,000 | 100 |
| SVHN | $32 \times 32$ | 73,257 | 26,032 | 10 |

## F.2 Parameter Information

Consistent training settings are maintained to ensure a fair comparison. For training input VR patterns, we apply the Adam optimizer [26] with an initial learning rate of 0.01. The number of epochs is 200, with the learning rate decay being 0.1, scheduled at epochs 100 and 145. All experiments are conducted on a single A100 GPU and the average accuracy of three distinct random seeds are reported.

# G Parameter Analysis

## G.1 Choosing Hyper-parameters

Table 5: Tuning ratio $\alpha$ and Laplace $\lambda$ (ResNet-18, Flowers102, average accuracy (%))

| $\alpha|\lambda$ | 0.01 | 0.1 | 1 | 10 | 100 | 1000 |
|---|---|---|---|---|---|---|
| 0.01 | 40.5±0.8 | 41.7±1.4 | 44.1±0.1 | 45.1±0.6 | 42.9±0.4 | 40.5±0.4 |
| 0.05 | 46.2±0.4 | 45.8±0.8 | 48.9±0.2 | 47.2±0.4 | 45.2±0.8 | 43.0±0.1 |
| 0.15 | 48.2±0.4 | 49.4±1.0 | 50.1±0.6 | 48.1±0.6 | 45.4±0.7 | 44.6±0.2 |
| 0.1 | 48.6±0.8 | 50.0±1.0 | 48.4±0.4 | 48.4±0.6 | 45.8±0.9 | 45.6±0.5 |
| 1 | 49.1±0.8 | 50.2±0.2 | 49.3±0.7 | 49.3±0.6 | 45.3±0.7 | 44.4±0.7 |
| Average | 46.5±0.6 | 47.4±0.9 | 48.1±0.4 | 47.6±0.5 | 44.9±0.7 | 43.6±0.4 |

As described in Section 4, the ratio $\alpha$ is used in calculating $k = \lfloor \alpha \cdot k_{\mathrm{T}} \rfloor$. The experimental results to tune hyper-parameters $\alpha$ and $\lambda$ are reported in Table 5. $\alpha$ is chosen among $[0.01, 0.05, 0.15, 0.5, 1]$, while $\lambda$ is chosen among $[0.01, 0.1, 1, 10, 100, 1000]$. The optimized $\lambda$ is determined first to be 1 by the average accuracy of different $\alpha$ values, followed by deriving an optimal $\alpha = 0.15$.

While the same hyper-parameters may not necessarily be optimal across different datasets, for the sake of consistency and fairness, this paper employs identical hyper-parameters for all datasets.

## G.2 Analyzing Hyper-parameters

Figures 9 and Figure 10 illustrate the impact of $\lambda$ and $\alpha$ on accuracy. It is observed that the optimal hyper-parameters vary across different datasets.

In general, as $\lambda$ increases, the test accuracy initially rises and then declines. This parameter is used to balance the contributions of individual pretrained labels. An over-small $\lambda$ might overly rely on the distribution of pretrained labels obtained from pretrained models, while a too-large one might overlook differences among pretrained labels. Meanwhile, with an increase in $\alpha$, accuracy first increases, then plateaus or slightly decreases. This is because excessively small or large $\alpha$ values may

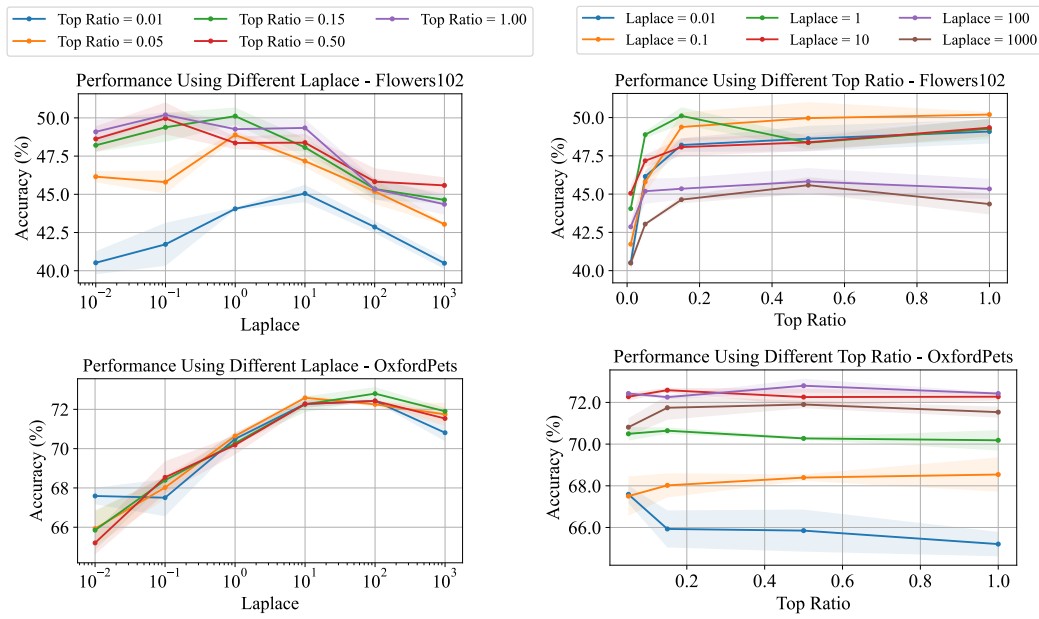

Figure 9: Accuracy with different Laplace $\lambda$.    Figure 10: Accuracy with different Ratio $\alpha$.

lead to the neglect of certain crucial labels or the emphasis on redundant ones during the estimation of the probability aggregation matrix. Therefore, choosing moderate values for $\lambda$ and $\alpha$ appears to be more appropriate.

### G.3    Task-specific Hyper-parameters

Table 6: Difference between task-specific parameters and shared parameters

|  | Flowers102 | UCF101 | DTD | OxfordPets | CIFAR10 |
|---|---|---|---|---|---|
| Specific $\alpha$ | 0.15 | 0.15 | 0.5 | 0.5 | 0.5 |
| Specific $\lambda$ | 1 | 1 | 1 | 10 | 10 |
| Accuracy (Trained on 70% Samples) | 45.82 | 31.84 | **43.75** | **72.27** | **66.54** |
| Shared $\alpha$ | 0.15 | 0.15 | 0.15 | 0.15 | 0.15 |
| Shared $\lambda$ | 1 | 1 | 1 | 1 | 1 |
| Accuracy (Trained on 70% Samples) | 45.82 | 31.84 | 42.31 | 70.52 | 66.04 |

We used universal hyper-parameters to show that BLM and BLM+'s performance gains over baselines are not sensitive to hyper-parameters. However, we assume that the dataset-specific tuning for hyper-parameters could yield more optimized results.

Additional experiments are conducted using a validation set and training set split of 30% and 70% to find optimal hyper-parameters for each dataset. Results are shown in Table 6. We observe that optimal hyper-parameters tailored for each dataset achieve better performance compared to using shared hyper-parameters, which matches our assumption.

## H    Limitations of BLM and BLM+

**Less effective for tasks with very few classes.** As shown in Table 1, when the number of classes (i.e., size of the label space) in downstream tasks is smaller (10 classes in SVHN and 10 classes in EuroSAT) and the original task is relatively simple, the advantage of BLM and BLM+ is not very pronounced. This is because BLM and BLM+ replace the one-to-one mapping with a pairwise-connected probabilistic LM. While this optimization yields positive results in most tasks, for a

small subset of simple tasks, the one-to-one mapping may better reflect the relationship between the pretrained label space and the downstream label space. For such tasks, BLM and BLM+ no longer exhibit significant effects.

**Not solving the cases where VR is not applicable for downstream tasks.** For example, in the case of the StanfordCars dataset in vision models, as shown in Table 1 and Table 2, the accuracy of the downstream task remains consistently low (<10%) through learning using input VR. While applying BLM and BLM+ in such scenarios yields better results compared to using one-to-one mapping, it still cannot significantly enhance VR performance to the extent of being comparable to finetuning the entire model.

# I   Visualization of Label Mapping Matrices

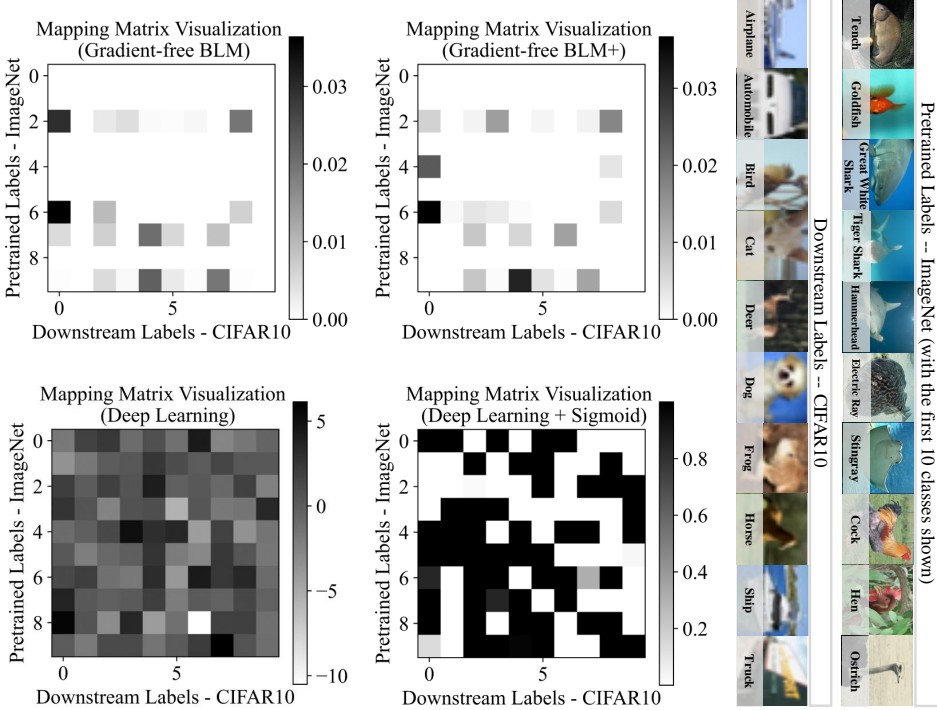

Figure 11: The visualization results of the LM matrices. Using the example of ResNet-18 pretrained on ImageNet-1K applied to the downstream task CIFAR10, the left figure displays the first 10 rows and 10 columns of the LM matrices (including the result matrix of the first 10 pretrained and downstream labels), while the right figure presents specific labels. Compared to gradient-free LM methods (i.e., BLM and BLM+), deep learning-based methods (i.e., a single-layer unrestricted neural network $\omega \in \mathbb{R}^{k_S \times k_T}$ and a single-layer neural network with Sigmoid $\omega \in [0, 1]^{k_S \times k_T}$) demonstrate less interpretability in revealing the relationship between labels.

Based on the example of ResNet-18 pretrained on ImageNet-1K applying to the downstream task CIFAR10, Figure 11 depicts the visualization results of LM matrices. The first row in Figure 11 shows the results of gradient-free methods BLM and BLM+, while the second row shows deep learning-based methods which learn a linear neural network for $f_{\text{out}}^{\omega}$. 'Deep Learning' refers to a single-layer neural network without constraints (i.e., $\omega \in \mathbb{R}^{k_S \times k_T}$), while 'Deep Learning + Sigmoid' refers to applying the Sigmoid function to restrict $\omega \in [0, 1]^{k_S \times k_T}$ aligning with the range of $\omega_{\text{BLM}}$ and $\omega_{\text{BLM+}}$. The right part of Figure 11 depicts the specific pretrained and downstream labels corresponding to these matrices.

It is observed that BLM and BLM+ are good at revealing similarities between pretrained and downstream labels. For example, for the downstream label 'Airplane', which visually resembles 'Great White Shark', 'Hammerhead' and 'Stingray', the weights in $\omega_{\text{BLM}}$ or $\omega_{\text{BLM+}}$ tend to be

higher. Conversely, for dissimilar labels like 'Truck' and 'Ostrich', the weights will be approaching 0. However, the weight matrices obtained from deep learning-based methods fail to capture such clear-label relationship. The results demonstrate the advantages of BLM and BLM+ in terms of interpretability.

## J  Training Cost Analysis

Table 7: Impact of epoch numbers on different label mapping methods

| | BLM | | | | BLM+ | | | | ILM | FLM |
|---|---|---|---|---|---|---|---|---|---|---|
| Epochs | 60 | 100 | 150 | 200 | 60 | 100 | 150 | 200 | 200 | 200 |
| Average Accuracy on 12 Tasks (%) | 44.5 | 45.2 | 45.5 | 45.3 | 45.8 | 46.4 | 46.9 | 46.7 | 40.6 | 37.2 |

Table 8: Training cost analysis of LM & VR and none-VR finetuning (on Flowers102)

| | Gradient-free LM | | | | | | Deep Learning-based LM | Finetuning | |
|---|---|---|---|---|---|---|---|---|---|
| | FLM | ILM | BLM | BLM+ | **BLM*** | **BLM+*** | - | Linear | Fully |
| Back-propagation when Learning LM | No | No | No | No | No | No | Yes | - | - |
| ResNet-18 | | | | | | | | | |
| Parameter Number (M) | 0.10 | 0.10 | 0.10 | 0.10 | 0.10 | 0.10 | 0.20 | 0.51 | 11.7 |
| Whole Time (min) | 11.97 | 12.04 | 11.95 | 13.06 | **6.03** | **6.52** | 12.34 | 14.03 | 15.28 |
| ResNeXt-101 | | | | | | | | | |
| Parameter Number (M) | 0.10 | 0.10 | 0.10 | 0.10 | 0.10 | 0.10 | 0.20 | 2.0 | 88.8 |
| Whole Time (min) | 24.68 | 24.81 | 24.51 | 24.71 | **12.33** | **12.44** | 24.80 | 24.49 | 35.07 |

**The Required Number of Epochs.** Different label mapping methods require varying numbers of epochs to converge. We initially used 200 epochs as with [4] to ensure a fair comparison with the baseline methods. Additional experiments are conducted to assess the impact of different epoch numbers from [60, 100, 150] on our BLM and BLM+ model, using ResNet-18 as the pretrained model. The results are shown in Table 7.

We found that running 100 epochs yields results comparable to those achieved with 200 epochs. This demonstrates that BLM and BLM+ require less convergence time, highlighting their efficiency.

**Overall Time Consumption.** Table 8 presents a comparison of different output mapping methods in terms of computational resources, utilizing the Flowers102 dataset as the downstream task. Gradient-free LM refers to estimating output mappings using statistical methods, while deep learning-based LM treats label mapping as a single linear layer neural network attached after the pretrained models. 'BLM*' and 'BLM+*' refer to training with only 100 epochs as is shown in Table 7. It should be noted that the running times for ILM, BLM, and BLM+ are measured using the quick version (see Appendix D.6 for details). Apart from VR methods, which fix the pretrained model, the time costs associated with directly finetuning pretrained models are also listed. Here, the term 'Linear' refers to finetuning the final layer of the pretrained model, while 'Fully' refers to finetuning the entire model.

Besides, regarding the performance of finetuning methods on downstream tasks compared with VR, please refer to [4] for more discussion. Since we mainly focus on LM methods for VR in this paper, which has a different problem setting with finetuning, the performance comparison of VR and finetuning will not be addressed here.

We therefore analyze the efficiency of BLM and BLM+ from three perspectives:

- Extra Consumption of Calculating the Mapping Matrix Compared with One-to-One Mapping: Compared to the baseline method ILM, the additional cost for BLM and BLM+ primarily involves the gradient-free multiplication and division within the mapping matrix (which is sized according to the source and target label spaces, $1000 \times 102$ in this case). This additional cost is minimal, as shown in Table 8.

- Time Consumption of Updating the Mapping Matrix per Epoch: Compared with FLM, updating $\omega$ in ILM, BLM, and BLM+ does not require running the model to obtain current predictions for each epoch. Instead, predictions from the most recent epoch can be reused (see Appendix D.6). As a result, there is no noticeable time overhead for updating $\omega$ per epoch, as indicated by Table 8.

- Time Consumption of LM and VR Compared with Deep Learning-based Methods: It is observed that methods based on deep learning introduce a substantial number of extra parameters (which would further increase with larger downstream label space and higher pretrained model complexity) along with the necessity of backpropagation for gradient computation. Conversely, the gradient-free LM methods along with VR emphasized in this study do not encounter these challenges.

# K  More Results on Visual Classification Tasks

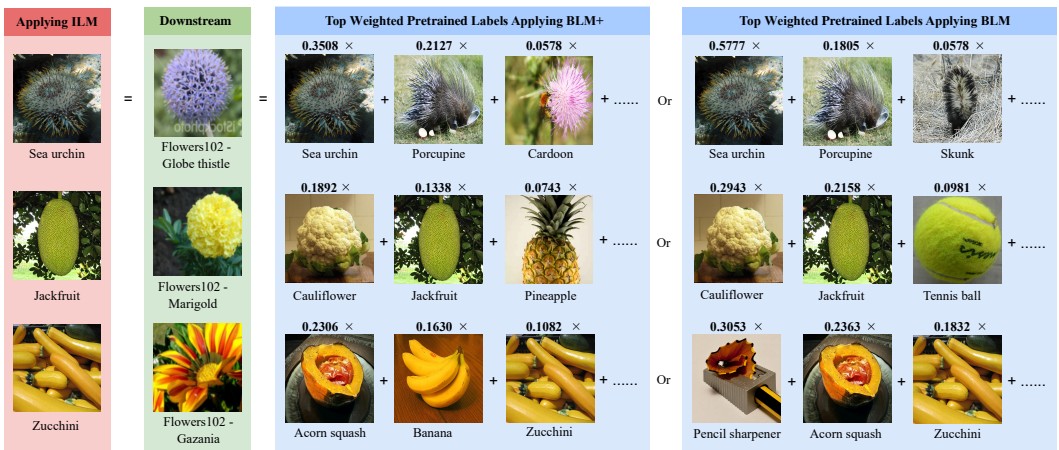

Figure 12: Label mapping results of ILM, BLM, BLM+ for VR on Flowers102 dataset.

Figures 12-16 illustrate the visualization results of label mapping using ILM (one-to-one mapping), BLM, and BLM+ for VR on various datasets with pretrained ResNet-18. For BLM and BLM+, the top three contributing pretrained labels corresponding to the downstream label are presented, along with their respective weights.

**Results when the pretrained and downstream labels exhibit appearance resemblance.** Figures 12 and 15 respectively depict the outcomes on Flowers102 and Food101 datasets, each about classification tasks of various flowers and food. BLM+ is adept at assigning higher weights to pretrained labels with a greater resemblance to the downstream label in terms of *color*, *shape*, and *intricate features*. In terms of *color*, as evidenced in Figure 15, the top-weighted labels for 'Edamame' comprise 'Green Snake', 'Artichoke', and 'Green Mamba', all sharing a green hue. Regarding *shape*, in Figure 12, the 'Gazania' with petal stripes corresponds to top weighted labels such as 'Banana' and 'Zucchini', which exhibit similar striping patterns. As for *intricate features*, in Figure 12, the 'Globe Thistle' with needle-like appearance aligns with top weighted labels including 'Sea Urchin', 'Porcupine', and 'Cardoon', which possess akin prickly characteristics.

**Results when the pretrained and downstream labels exhibit similarities in texture.** Figure 13 presents the results on the DTD dataset, which pertains to the classification of various textures. Both BLM+ and BLM assign higher weights to labels sharing akin textures. For example, 'Spiralled'

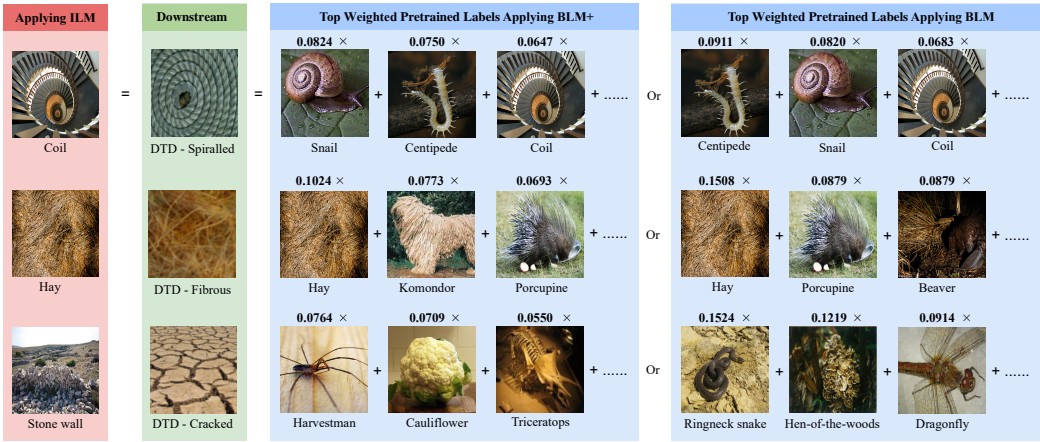

Figure 13: Label mapping results of ILM, BLM, BLM+ for VR on DTD dataset.

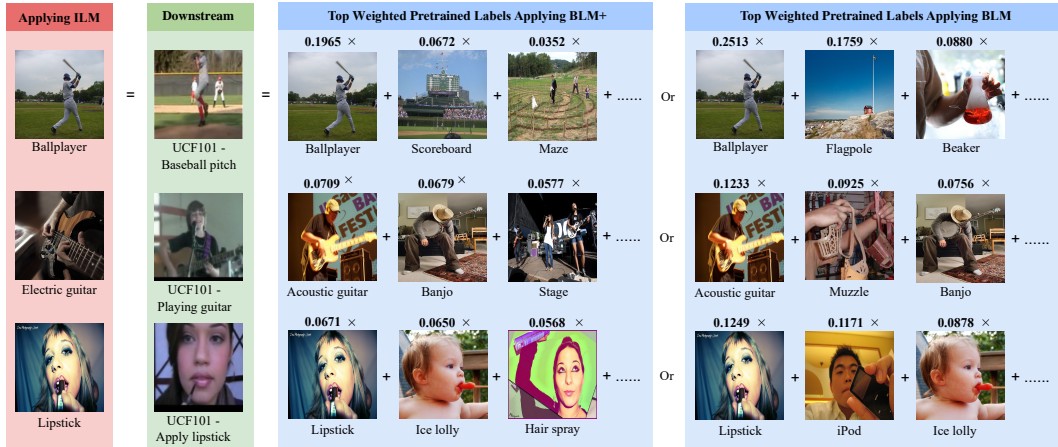

Figure 14: Label mapping results of ILM, BLM, BLM+ for VR on UCF101 dataset.

corresponds to top-weighted labels embodying spiral-shaped entities such as 'Snail', 'Centipede', and 'Coil', while 'Fibrous' aligns with entities possessing a rough and fibrous texture, including 'Hay', 'Komondor', and 'Porcupine'.

**Results when the pretrained and downstream labels exhibit similarities in backgrounds.** Figure 14 illustrates the results on the UCF101 dataset, a dataset for action classification. In this task, both BLM and BLM+ tend to assign higher weights to pretrained labels with backgrounds or environments akin to the downstream labels. For example, the action 'Apply Lipstick' often involves the presence of a human face; hence, pretrained labels such as applying 'Lipstick', eating 'Ice Lolly', and spraying 'Hair Spray' contribute significantly. Likewise, labels closely associated with 'Baseball pitch' include 'Ballplayer' and 'Scoreboard', featuring backgrounds of vast grass fields.

**Results when pretrained and downstream labels exhibit inclusion relationship.** Figure 16 illustrates the results on the CIFAR10 dataset, which comprises images broadly categorized into ten main classes, with each category corresponding to several subcategories within the pretrained domain. It is noted that unlike the singular class selection of ILM, both BLM and BLM+ allocate similar weights to multiple subcategories. For example, 'Dog' corresponds to different breeds such as 'Cocker Spaniel', 'English Springer', and 'English Setter', while 'Bird' encompasses subcategories including 'Peacock', 'Albatross', and 'Little Blue Heron'. Hence, the learning framework of BLM and BLM+ demonstrates effective handling of the inclusion relationship between label spaces.

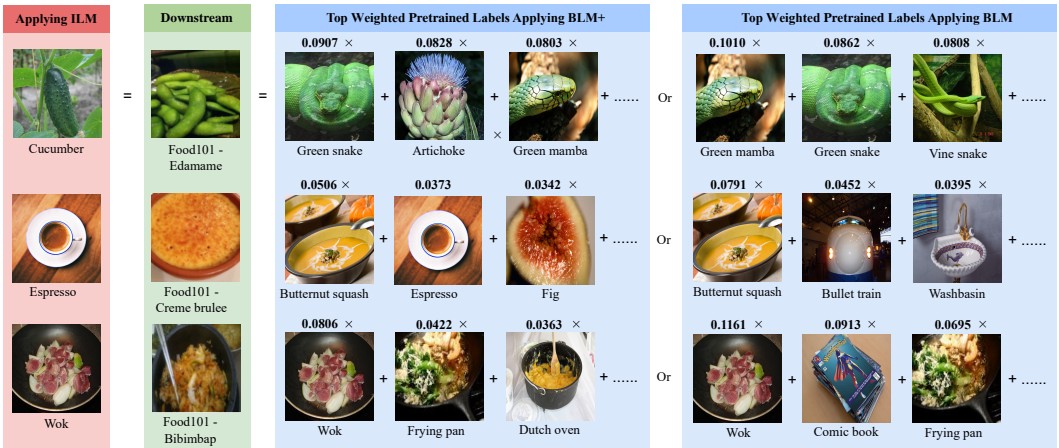

Figure 15: Label mapping results of ILM, BLM, BLM+ for VR on Food101 dataset.

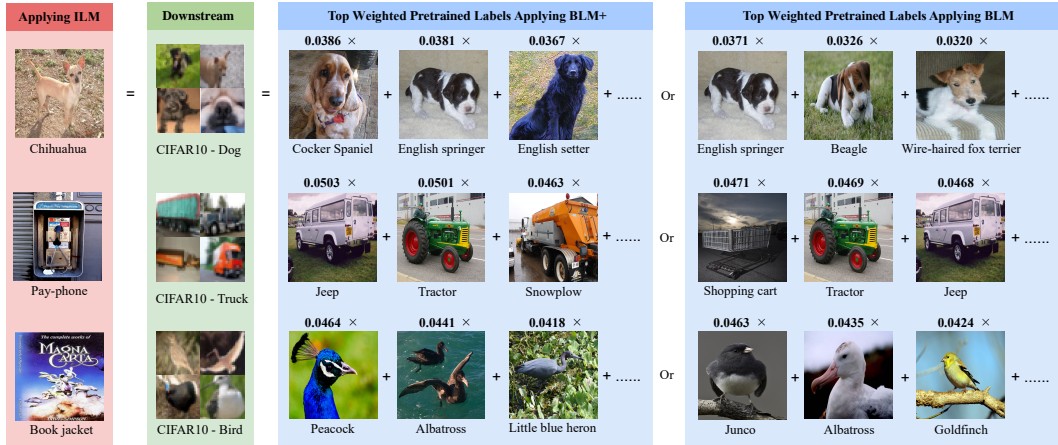

Figure 16: Label mapping results of ILM, BLM, BLM+ for VR on CIFAR10 dataset.

# L  Applications on Vision-Language Models

## L.1  Learning Framework

The distinction between *Vision-Language Models* (VLM) and vision models lies in (1) vision models take a single image as input, whereas VLMs take a pair of text and images as input; and (2) vision models have fixed pretrained labels, with model outputs being logits, while VLMs lack pretrained labels, with model outputs being the cosine similarity [57] between images and text embeddings. As a result, when applying BLM and BLM+ to VLM, it is necessary to design an input text set to replace the original pretrained labels in vision models.

In this paper, we follow a previous work [4] and construct the input texts set by taking the Cartesian product [56] of the downstream label set and the prompt set. BLM and BLM+ can be applied to compute the joint frequency distribution (for BLM) or aggregated predicted probability (for BLM+) between the input texts and the downstream ground-truth labels. This enables the mapping from candidate input texts to probable classification results.

The full process of learning $\omega_{\text{BLM}}, \omega_{\text{BLM}+}$ for vision models or VLMs is illustrated in Figure 17. Besides, the pipeline and algorithm details are the same as BLM and BLM+ for vision models shown in Figure 2, Algorithm 7 and 9.

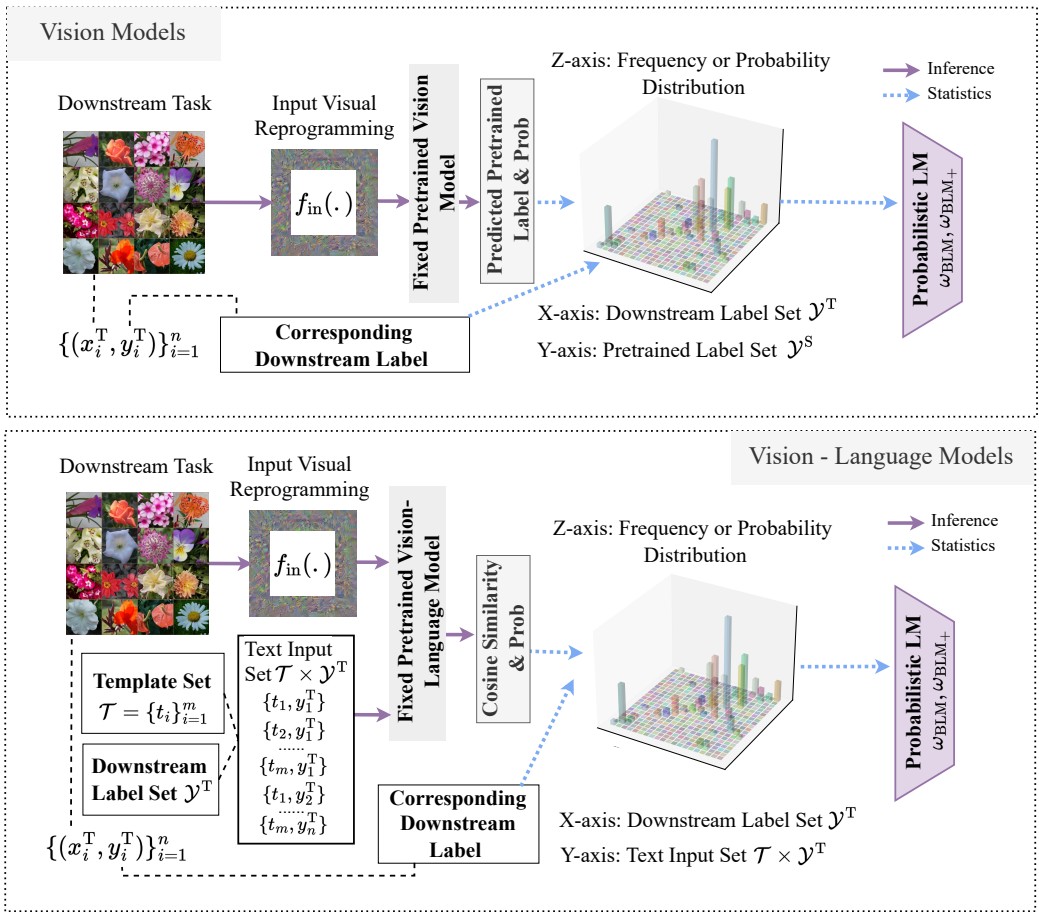

Figure 17: The framework of learning $\omega_{\text{BLM}}$ or $\omega_{\text{BLM+}}$ for pretrained vision models (upper) or VLMs (lower). As described in Section 4, for vision models, $\omega_{\text{BLM}}$ or $\omega_{\text{BLM+}}$ is derived from the frequency distribution (in BLM) or probability aggregation matrix (in BLM+) where pairs of [predicted pretrained label, ground-truth downstream label] are calculated. Nevertheless, for VLMs, the predicted pretrained label is replaced by possible text inputs from the Cartesian product of the downstream label set, and the prompt set. The cosine similarities of images and text embedding are calculated in VLMs to replace the output logits in vision models.

## L.2 Performance Results

Table 9 presents the performance of BLM and BLM+ applied to VLMs across 12 datasets. For a fair comparison, we follow the previous work [1] to employ CLIP as the pretrained model and a watermarking-based VR with an outer frame size of 30. We utilized an initial learning rate of 40 and a Cosine Annealing learning rate schedule [35], with a total of 200 epochs. An SGD optimizer with a momentum of 0.9 was employed for learning the Input VR. Results without label mapping (denoted by 'None') and one-to-one mapping served as the baseline, and the average accuracy was computed over three different random seeds.

From the performance results, it can be observed that except for the EuroSAT dataset, which has a small number of classes and simpler tasks (this limitation will be discussed in detail in Appendix H), BLM or BLM+ achieves improvements across all other tasks. They achieve the average accuracy of 79.1% and 79.3%, respectively, without increasing the number of model parameters to be trained. This empirical evidence demonstrates that BLM and BLM+ can also be effectively applied to VLMs.

Table 9: Performance comparison on VLMs (mean % ± std %)

| CLIP (ViT-B32) | Baseline | | Ours | |
| --- | --- | --- | --- | --- |
| Method | None | One-to-one Mapping | BLM | BLM+ |
| Flowers102 | 70.5±0.7 | 75.5±1.0 | **76.9**±1.9 | 76.4±1.5 |
| DTD | 61.4±0.6 | 59.5±1.1 | 60.9±0.9 | **61.5**±0.3 |
| UCF101 | 67.5±0.1 | 67.9±0.6 | 72.2±0.2 | **72.3**±0.4 |
| Food101 | 79.2±0.2 | 78.1±0.3 | 79.3±0.1 | **79.4**±0.1 |
| GTSRB | 91.4±0.4 | 91.3±0.2 | **91.5**±0.2 | 90.9±1.0 |
| EuroSAT | **96.6**±0.1 | 96.5±0.1 | 96.3±0.1 | 96.3±0.1 |
| OxfordPets | 88.4±0.1 | 86.8±0.6 | 88.6±0.5 | **89.0**±0.4 |
| StanfordCars | 57.9±0.1 | 55.8±0.1 | 59.8±0.7 | **60.3**±0.2 |
| SUN397 | 61.4±0.2 | 60.6±0.1 | 63.1±0.2 | **63.8**±0.2 |
| CIFAR10 | 94.0±0.2 | 94.1±0.1 | **94.2**±0.2 | 94.1±0.3 |
| CIFAR100 | 75.1±0.2 | 74.8±0.1 | 75.4±0.5 | **75.5**±0.3 |
| SVHN | 91.3±0.2 | 91.3±0.2 | 91.5±0.1 | **91.7**±0.1 |
| Average | 77.9 | 77.7 | 79.1 | **79.3** |

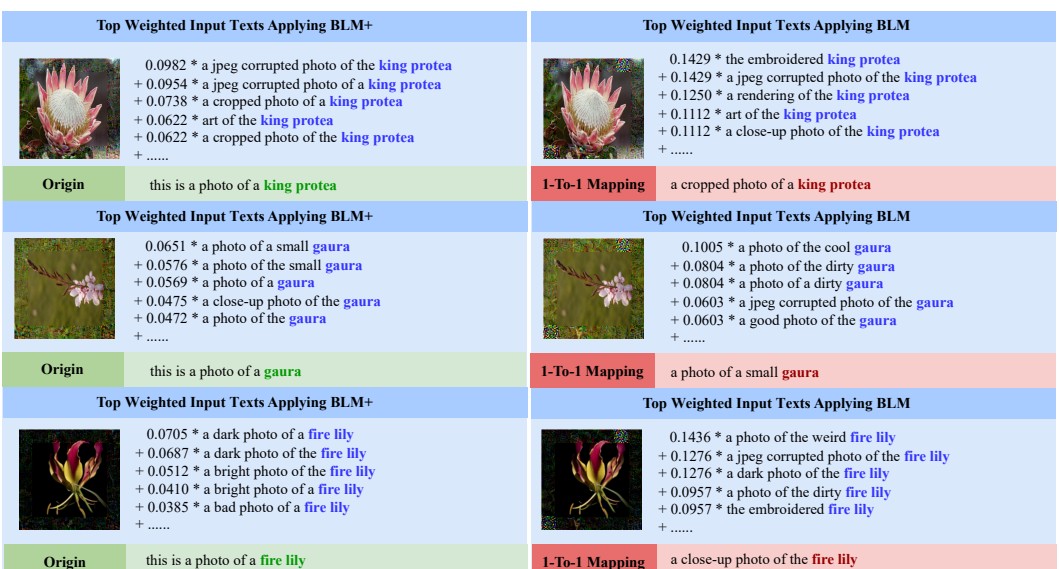

Figure 18: Results of ILM, BLM, BLM+ for VR on Flowers102 dataset.

## L.3 Visualization Results

Figures 18-22 show the visualization results of top-weighted input texts on different datasets applying BLM and BLM+. It is evident that, unlike the single optimal text input in one-to-one mapping, BLM and BLM+ assign different weights to many possible descriptions. For example, in CIFAR10, an image of a bird may be described as 'a low-resolution photo of a bird', 'a close-up photo of the bird', or 'this is a photo of a bird', among others. Such methods affirm different expressions instead of only one description using one-to-one LM.

These experiments further demonstrate that BLM and BLM+ can be used to enhance the performance of input VR in VLMs while providing reasonable explanations for why input VR in VLMs can effectively work.

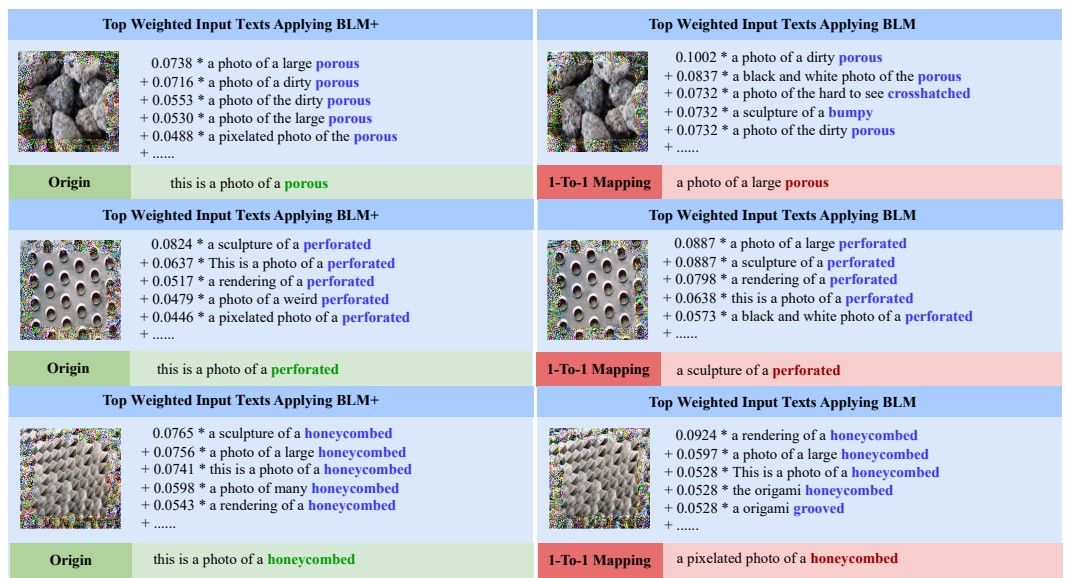

Figure 19: Results of ILM, BLM, BLM+ for VR based on CLIP on DTD Dataset.

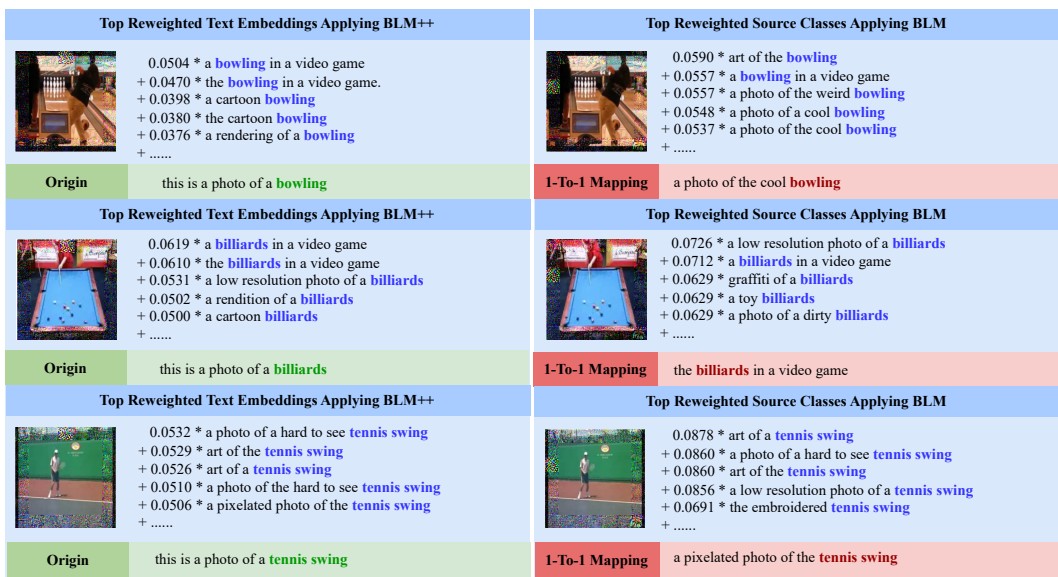

Figure 20: Results of ILM, BLM, BLM+ for VR based on CLIP on UCF101 dataset.

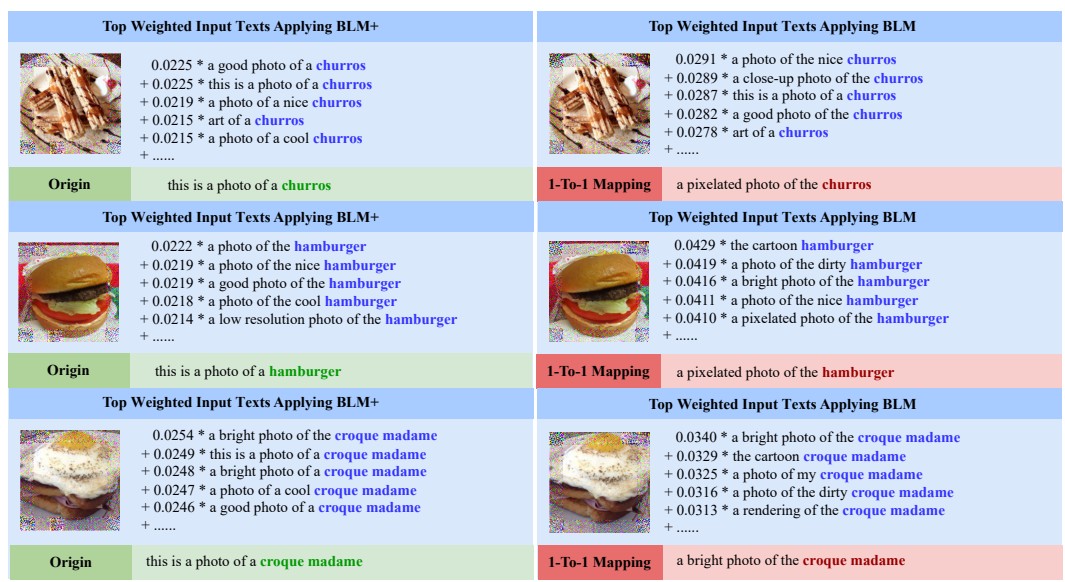

Figure 21: Results of ILM, BLM, BLM+ for VR based on CLIP on Food101 dataset.

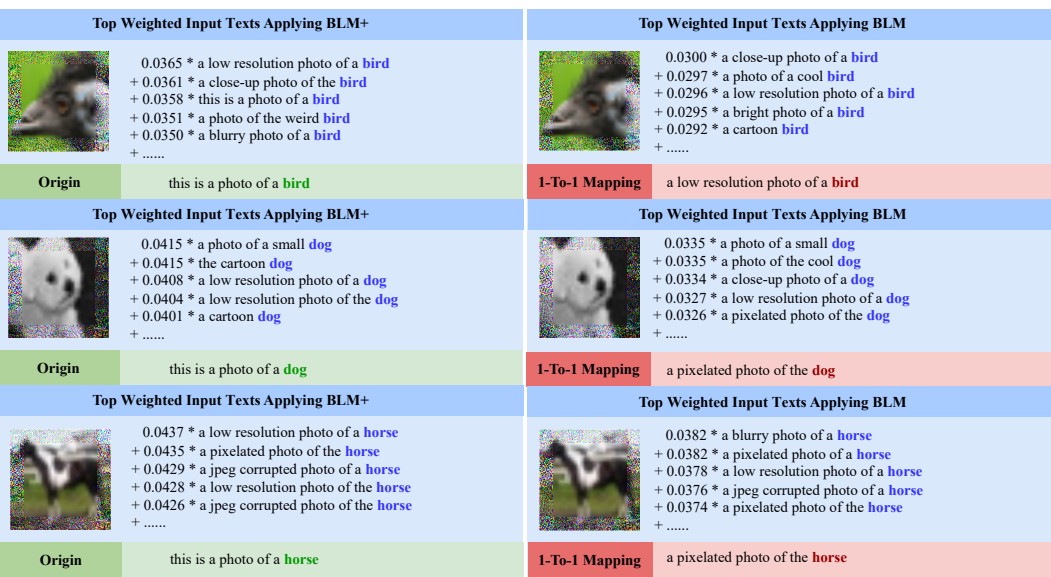

Figure 22: Results of ILM, BLM, BLM+ for VR based on CLIP on CIFAR10 dataset.

