# OpenReview forum: "Bayesian-guided Label Mapping for Visual Reprogramming"
_NeurIPS.cc/2024/Conference — NeurIPS 2024 oral_

### Official Review · Reviewer_iW2K · 2024-07-10

**Soundness:** 4
**Presentation:** 4
**Contribution:** 4
**Rating:** 7
**Confidence:** 4

**Summary:**

This paper focuses on a technique called label mapping (LM) which is used in visual reprogramming. It finds the relation between pre-trained labels and downstream labels. In conventional fine-tuning, LM is just the backpropogation using the loss function. However, this paper and previous similar papers aim to develop a gradient-free way to find the weights in the last layer in the conventional fine-tuning. The whole paper is based on Bayes' rules (using it on the output of a pre-trained model, which is interesting), which is solid and theoretically justified.

**Strengths:**

1. Good paper flow. The flow of this paper is good and appreciated. Directly analyzing the loss using the Bayes' rule is clear.

2. Extensive experiments. Except for the good performance, the authors also try to understand the good performance, which is interesting. Especially on the visualization of top relationship between pre-trained labels and downstream labels. The result is convincing, as the selected top pre-trained labels are indeed related to downstream tasks.

3. Timing topic. I can see some applications which might need this kind of techniques. Keeping the integrity of a pre-trained model has many advantages, e.g., not lose the generalization of the pre-trained model.

4. Solid analysis. Two theoretical contributions are included in this paper. One is how to embed a pre-trained model into p(y^T|x^T), another one is how BLM has a higher accuracy.

**Weaknesses:**

1. Benchmark datasets follow the previous papers, which is good. However, these tasks seem not the most relevant to what you propose in this paper. Are there some scenarios where the fine-tuning the last layer is not feasible?

2. The paragraph between 253-266 is confusing, which can be moved to other section. Appendix perhaps.

3. One major drawback of the presentation is the algorithm tables. The equations are still not the best way for practitioners. Main algorithm table should be moved in Section 4. After looking at the algorithms, it is easy to find that the proposed method is easy to be implemented. However, I cannot feel that after I read the section 4.

4. How to choose Padding or Watermarking in practice?

5. Experiments to increase n are not necessary. Could the authors explain why it matters? As long as the method is valid, the accuracy will be improved when n increase.

**Questions:**

See the weaknesses above.

**Limitations:**

No concerns for this paper.

---

> ### Author Rebuttal · Authors · 2024-08-07
>
> **W1:** Thank you for your question. This paper falls in the scope of Visual Reprogramming (VR). Thus, the experimental setting of this paper aligns with those used in previous VR studies, particularly ILM [1], to ensure comparability and consistency in the field. This setting serves as an established benchmark for evaluating VR techniques.
>
> Regarding when fine-tuning the last layer is not feasible, we consider the following cases:
> 1. copyright and legal constraints: modifying some pretrained models may violate licensing agreements or intellectual property rights.
> 2. prevent catastrophic forgetting: keeping the pretrained model intact helps to maintain its general knowledge across tasks
> 3. black-box optimization: in addition, our method is useful when only the predicted logits of input are available. It does not need access to the model's internal parameters which fine-tuning clearly struggles to handle.
>
> **W2:** Thank you for your question. Due to space limitations, lines 253-266 of the original text are not fully described. We will move this section to the appendix in the next version and include detailed information.
>
> **W3:** We appreciate your feedback regarding the presentation of Sec. 4 and algorithm tables. We acknowledge your point about the placement of the algorithm tables. Due to space constraints, we had initially placed them in the Appendix (lines 455 and 462). However, we recognize the importance of making these algorithms more readily accessible to readers. We plan to move the main algorithm table to Sec. 4 as you suggested in the final version.
>
> We understand your concern regarding the equations in Sec. 4. While the mathematical formulations are important for a rigorous presentation, we will smooth out the section with more practical insights. We aim to strike a balance between mathematical precision and practical applicability.
>
> **W4:** Thank you for your question. The use of Padding or Watermarking depends on the specific downstream tasks, and we believe the selection may partly depend on the relationship between the pretrained model's input size and the downstream task's image dimensions.
> For instance, let’s consider the results of BLM+ using ResNet-18 as the pre-trained model:
>
> | | GTSRB | SVHN | CIFAR10 | CIFAR100 | Flowers102 | EuroSAT | OxfordPets | SUN397 | UCF101 | Food101 | DTD |
> | -------- | -------- | -------- | -------- | -------- | -------- | ------- | -------- | -------- | -------- | -------- | -------- |
> | Image Size | 32 | 32 | 32 | 32 | 128 | 128 | 128 | 128 | 128 | 128 | 128 |
> | Padding | 54.3 | 74.2 | 66.8 | 30.6 | 50.1 | 86.7 | 70.6 | 18.7 | 32.0 | 25.1 | 43.9 |
> | Watermarking | 82.0 | 78.8 | 75.7 | 41.6 | 44.1 | 84.8 | 73.3 | 19.4 | 35.4 | 22.9 | 43.0 |
>
> Our observations:
> - watermarking performs better when there is a significant size disparity: for tasks with much smaller images (32x32, e.g., GTSRB, SVHN, CIFAR10, CIFAR100) compared to the pre-trained model's input (224x224), watermarking often outperforms padding. It prevents introducing too many parameters around the image and significantly downscaling it.
> - padding may perform better with larger downstream images in some cases: for tasks using larger images (128x128, e.g., Flowers102, EuroSAT), padding tends to perform better. It maintains the original image integrity by avoiding pixel value alterations.
>
> In general, the choice impacts how we adapt pretrained model to new tasks. Therefore, the goal for input VR is to maximize the transfer of learned features while accommodating the new task's visual characteristics - the optimal choice may depend on factors beyond just image size and deserves future explorations.
>
> **W5:** Thank you! Indeed, the accuracy typically improves with larger training sets. Our focus on smaller $n$ values serves as a purpose in evaluating the robustness of our proposed methods, BLM/BLM+, compared to the baseline ILM under limited data availability conditions. Our rationale for this evaluation is:
> 1. Practical implications: VR was proposed for adapting pretrained model to data-limited tasks. In this sense, obtaining large-scale labeled downstream task data can be challenging. By demonstrating robust performance with smaller $n$ (25%, 50%, and 75% of the original size), we highlight the practical advantages of BLM/BLM+ in data-limited scenarios, which better aligns with real-world applications.
> 2. Overfitting risk: Smaller training sets inherently carry a higher risk of overfitting. By testing our methods under these conditions, we can better assess their generalization capabilities compared to the baseline ILM.
> Thus, these experiments were conducted to validate the effectiveness and reliability of BLM/BLM+ across various data regimes. While increasing $n$ may not be that meaningful, our evaluation aims to demonstrate that BLM/BLM+ maintains competitive performance even with limited data, thus providing a more comprehensive evaluation of the practical utility.
>
> [1] Chen et al. Understanding and improving visual prompting: A label-mapping perspective. In CVPR, 2023.

---

### Official Review · Reviewer_zxuh · 2024-07-11

**Soundness:** 3
**Presentation:** 3
**Contribution:** 3
**Rating:** 8
**Confidence:** 4

**Summary:**

Visual reprogramming is an interesting way to reuse a pre-trained classifier or an VLM. In previous methods, the way to change the output interface is basically gradient-free and one-on-one mapping. In this paper, the authors found that the previous way is suboptimal and ignores  information. Then, from a theoretical perspective, a new objective is derived using Bayes' theorem. By optimizing this objective, the performance of VR methods is significantly improved, which is demonstrated via extensive experiments. A theoretical study is provided as well.

**Strengths:**

1. The research topic is of significance. VR is useful in practice. This paper understands and extends the label mapping from a mathematical perspective, which steps much further compared to previous methods.

2. The derivation regarding w is could generally cover previous methods (which is probably the first). The drawback of previous methods is manifested based on the derivation.

3. Math expressions are easy to follow.

4. Abundant experiments are provided, including large datasets, large models, and VLM. From the experimental results, we can see the improvement is universal, justifying the general effectiveness of the proposed method.

**Weaknesses:**

1. In Eq. (1), what is the real gap between right and left terms? It seems not easy to get this gap. I suggest using right one directly, which is easy to be acceptable.

2. Above the Eq. (6), one might misunderstand the way to calculate the frequency. Using () might be better.

3. Figure 2 is confusing. It looks like step 3 follows step 1, rather than step 2 following step 1. What is the actual sequence between step 1~3?

4. In section 4.2, the calculation of accuracy is confusing. $x$ is missing.

5. What does $\bar{r}$ mean in l.262?

6. Training time is potentially a concern. How many training epochs are needed for BLM?

**Questions:**

Please refer to "Weaknesses".

**Limitations:**

No further concerns regarding limitations.

---

> ### Author Rebuttal · Authors · 2024-08-07
>
> **W1:** Thanks! The gap between left-hand side (LHS) and right-hand side (RHS) comes from the relationship between Maximum Likelihood Estimation (LHS) and Empirical Risk Minimization (RHS) in statistical learning theory.
> - LHS: represents the true objective of VR is to maximize the conditional probability of downstream label given downstream input image.
> - RHS: provides a practical and empirical approximation using a finite training set and a chosen loss function.
>
> This formulation aligns with fundamental principles in learning theory:
> - We use empirical risk as a proxy for the expected risk.
> - A loss function is chosen to approximate the negative log-likelihood.
> - This approximation connects to generalization bounds. The gap between the expected and empirical risks can be bounded using techniques like VC-dimension.
> - The law of large numbers ensures convergence of empirical to expected risk as the training set size increases.
>
> However, we agree that using RHS directly is more straightforward. We plan to incorporate your suggestion in the next version for easier understanding.
>
> **W2:** Thanks for the suggestion, we will include () in the next version.
>
> **W3:** Thank you for your question. In fact, step 3 (applying $\omega$) is performed after step 2 (calculating $\omega$, and the blue dotted line indicates the flow of $\omega$). However, after calculating $\omega$, the logits output from step 1 will also be used to obtain the final prediction result in step 3 (as shown by the purple solid line). We will revise Fig. 2 to more clearly illustrate this process in the next version of the paper.
>
> **W4:** Thanks for pointing it out. The omission of $x$ in Eq. 10 was intentional because:
> - Our analysis in Sec 4.2 (and Appendix Sec. C) focuses on output label mapping (LM) and specifically compares different LM methods, provided that the pretrained model $f_{\rm pre}$, input $x$, and input transformations $f_{\rm in}$ are the same across different LM methods (details in Appendix Sec. C.1).
> - The expected accuracy calculation is based on conditional probabilities where $x$ is implicitly part of the condition. As all LM methods operate on the same $x$, we can safely omit it, allowing us to concisely highlight the differences of LM themselves.
>
> We will also add a note in the main text to clarify this omission, emphasizing our focus on comparing LM methods and the scope of this analysis.
>
> **W5:** Thank you for your question. We will clarify it in the next version. $y_r^{\rm S}$ means a class that is more relevant to $y_{i}^{\rm T}$ (line 256), while $y_{\bar r}^{\rm S}$ is a class that is less relevant to $y_{i}^{\rm T}$ (line 257). Here we assume two classes, $y_{r}^{\rm S}$ and $y_{\bar r}^{\rm S}$, in the output space of the pre-trained model, which satisfy $p(Y^{\rm T}=y_i^{\rm T}|Y^{\rm S}=y_r^{\rm S}, X^{\rm T})>p(Y^{\rm T}=y_i^{\rm T}|Y^{\rm S}=y_{\bar r}^{\rm S}, X^{\rm T})$.
>
> **W6:** Thank you for your question. Please refer to our reply to **Common Question 1**.

---

> > ### Comment · Reviewer_zxuh · 2024-08-11
> > **Response to Rebuttal**
> >
> > Thank you very much for your review. My concerns are addressed, and I have raised my score.

---

> > > ### Author Response · Authors · 2024-08-12
> > > **Thanks for your support!**
> > >
> > > Dear Reviewer zxuh,
> > >
> > > Many thanks for your support and increasing your score to 8! We will merge all comments into the updated version.
> > >
> > > Best regards,
> > >
> > > Authors of Submission7993

---

### Official Review · Reviewer_885t · 2024-07-12

**Soundness:** 4
**Presentation:** 3
**Contribution:** 4
**Rating:** 7
**Confidence:** 4

**Summary:**

Another type of transfer learning approach is considered in this paper: model reprogramming. Different from classical transfer learning approaches, model reprogramming only changes models via changing the input space and output space, which is more efficient to fit a pretrained model to some downstream tasks. Specifically, this paper focuses on the output space, label mapping. The motivation is strong as it is clear that single map does not work well. Then, new methods are proposed based on Bayes' theorem. Experiments are enough and solid, in terms of model size or dataset size.

**Strengths:**

1. Transfer learning is even more important in the current era. This paper focuses on an interesting direction, which is significant to the field.

2. The main result of this paper looks general, compared to previous label mapping methods. It is novel to first consider a pretrained model's output into the P(y|x) and then analyze what we should do.

3. I enjoy reading the experiment section, which is quite comprehensive. I did not see some necessary experiment missing.

**Weaknesses:**

1. Transfer learning literature is missing in this paper. Although model reprogramming is not considered as a transfer learning method in the literature, it is one of transfer learning techniques. This review can help position this paper better.

2. Can we transfer a pretrained model to any possible task? Are there standards to help choose which model should be used for a specific downstream task?

3. For example, if we can use a model trained with cars to help recognize images of animals? It looks impossible. How to tell when we can use a pretrained model?

4. Computing w in BLM also needs time. Is there efficient advantage of VR anymore? What is the performance of finetuning? The gap should be reported in this paper.

5. Main algorithm table should be moved to the main paper. It is not very easy to implement BLM or BLM+ based on the formula, but the algorithms are very helpful.

6. It would be better to explain meaning of some conditional probabilities.

**Questions:**

1. Transfer learning literature is missing in this paper. Although model reprogramming is not considered as a transfer learning method in the literature, it is one of transfer learning techniques. This review can help position this paper better.

2. Can we transfer a pretrained model to any possible task? Are there standards to help choose which model should be used for a specific downstream task?

3. For example, if we can use a model trained with cars to help recognize images of animals? It looks impossible. How to tell when we can use a pretrained model?

4. Computing w in BLM also needs time. Is there efficient advantage of VR anymore? What is the performance of finetuning? The gap should be reported in this paper.

**Limitations:**

No concerns for this paper.

---

> ### Author Rebuttal · Authors · 2024-08-07
>
> **W1/Q1:** Thank you very much for your suggestion. In the next version, we will expand our literature review to include relevant transfer learning concepts, particularly focusing on how Visual Reprogramming relates to and differs from traditional transfer learning methods. To better position this paper, we'll also discuss the spectrum of transfer learning techniques, connect and compare Visual Reprogramming with other parameter fine-tuning methods, highlighting its merits in scenarios where pretrained model preservation is crucial.
>
> **W2/Q2/W3/Q3:** Thanks for the question. To make our reply relevant, we will discuss the transferability in the context of VR.
>
> **Transferability in VR** Drawing on theoretical foundations from [1], the transferability of a pretrained model to a downstream task can be bounded by:
> $$\mathcal{L}^{\rm{T}} \leq \mathcal{L}^{\rm{S}} + \mathcal{W}(\mu^\rm{S}, \mu^\rm{T}) + \epsilon$$
> where $\mathcal{L}^{\rm{T}}$ and $\mathcal{L}^{\rm{S}}$ denotes error for downstream and pretrained tasks. $\mathcal{W}(\mu^\rm{S}, \mu^\rm{T})$ is the Wasserstein distance between the logit distributions of pretrained input $\mu^\rm{S} = f_{\rm pre} (x^{\rm S})$ and the reprogrammed downstream input $\mu^\rm{T} = (f_{\rm pre} \circ f_{\text{in}}) (x^{\rm T})$, $\epsilon$ is a small constant. We therefore know that the performance on downstream tasks can be related to the pretrained task performance and the alignment between pretrained task and downstream task.
>
> **Model selection** This bound suggests some insights on selecting models:
> - Pretrained model with higher capacity may lead to lower $\mathcal{L}^{\rm{S}}$,
> - Pretrained feature is relevant to the downstream task; or input VR and the output LM that effectively transform downstream input and output (both potentially contribute to lower $\mathcal{W}(\mu^\rm{S}, \mu^\rm{T})$ distance, indicating better alignment)
>
> For example, a model with large capacity pretrained on general object recognition (e.g., ResNet on ImageNet) may transfer better to another recognition task than a model pretrained on a highly specialized dataset.
>
> **Transferring between dissimilar domains (e.g., car-animal)** The feasibility depends on how is the Wasserstein distance minimized - this is theoretically possible even for seemingly unrelated domains, but the practical difficulty varies because of the challenge of measuring domain similarity and feature relevance.
>
> **When to use a pretrained model** Currently, there lacks theoretical tools to accurately measure the above Wasserstein distance BEFORE training, which makes it difficult to directly tell when and whether a pretrained model can be used. Therefore, we look forward to future work that focuses on effective ad-hoc estimation of such distance and techniques that minimize it through optimized VR and LM methods.
>
> In short, while VR offers a flexible way of transfer learning, the choice of pretrained model and its adaptability to a specific downstream task depends on both model capacity and the ability to align the feature distributions.
>
> **W4/Q4:** Thank you for your questions. For details on training time, please refer to **Common Question 1**. Regarding the fine-tuning results, since we follow the settings outlined in [2], we will directly quote these results from [2] in the next version of our paper.
>
> **W5:** Thank you for your suggestion. Due to space limitations, we have placed the algorithm tables in the appendix (lines 455 and 462). In the next version, we will prioritize incorporating them into the main text to enhance readability and understanding for our readers.
>
> **W6:** Thank you for your suggestion. We have added a detailed explanation in **Common Questions 2** and introduced a simple example to illustrate. We will include this revision in the final version.
>
>
> [1] Yang et al. Voice2Series: Reprogramming Acoustic Models for Time Series Classification. in ICML 2021.
>
> [2] Chen et al. Understanding and improving visual prompting: A label-mapping perspective. In CVPR, 2023.

---

> > ### Comment · Reviewer_885t · 2024-08-13
> >
> > My concerns have been addressed. I will support this paper with the positive score.

---

> > > ### Author Response · Authors · 2024-08-13
> > > **Many thanks for your support!**
> > >
> > > Dear Reviewer 885t,
> > >
> > > We are glad to hear that your concerns are addressed. Thanks for your support.
> > >
> > > Best,
> > >
> > > Authors of Submission7993

---

### Official Review · Reviewer_RgTa · 2024-07-14

**Soundness:** 4
**Presentation:** 3
**Contribution:** 3
**Rating:** 7
**Confidence:** 4

**Summary:**

Pretrained models play a crucial role in current machine learning and computer vision tasks, and effectively leveraging them in downstream tasks has become increasingly important. This paper explores the research area of visual reprogramming (VR), which diverges from traditional fine-tuning by adjusting the input space rather than the parameter space of pretrained models. Additionally, VR necessitates establishing a mapping from pretrained labels to downstream labels, which is the primary focus of this paper. The proposed method for mapping pretrained labels to downstream labels is well-motivated and convincing. The experimental results are robust and demonstrate the efficacy of the approach.

**Strengths:**

1. The experiments are solid and impressive, covering larger models compared to previous works in this field.

2. The motivation is very clear, adhering to basic probability rules.

3. Two methods are proposed based on different estimation approaches for the key component, demonstrating sufficient technical contribution.

4. Some experiments provide insightful explanations on why the mapping learned from the proposed method is effective, which I found particularly interesting.

**Weaknesses:**

1. Although the motivation is clear, it feels somewhat dry. More detailed explanations are needed for Section 4.1. Specifically, some of the conditional probabilities require further clarification (e.g., Eq. (6)).

2. It is unclear how Section 4.2 can be extended to the multi-class case. The accuracy metric (Acc) used here appears to be suited only for binary cases.

3. While it is commendable that a universal hyperparameter performs well in your experiments, it raises the question of whether these hyperparameters can be tuned using a validation set. How do all methods perform when evaluated on the same validation set?

4. I am not entirely convinced by Figure 4. Why is pineapple selected as well? This needs further explanation.

5. The absence of algorithm tables in the main content is a significant omission and should be addressed. Similar to the first weakness, the current method presentation feels somewhat dry.

6. Is BLM+ without Top-K and Bayes the second column in Table 3? If so, additional descriptions should be included for clarity.

7. What are the fine-tuning results in this context? It is essential to establish a standard to determine when VR is needed. Additionally, what is the total time cost compared to fine-tuning?

**Questions:**

See Weaknesses.

**Limitations:**

The authors have adequately discussed the limitations of the work.

---

> ### Author Rebuttal · Authors · 2024-08-07
>
> **W1:** Thank you. We have added a detailed explanation in **Common Question 2** and provided a simple example to illustrate the conditional probabilities.
>
> **W2:**  Thanks. We want to clarify how our analysis can be extended to multi-class cases.
>
> **Expected Accuracy Definition** The formula (Eq. 10) evaluates the expected probability of correctly mapping each $y^{\rm S} \in \mathcal{Y}^{\rm S}$ to the corresponding $y^{\rm T} \in \mathcal{Y}^{\rm T}$, remaining valid even for multi-class settings. It is agnostic to the number of classes in both $\mathcal{Y}^{\rm S}$ and $\mathcal{Y}^{\rm T}$.
>
> **Mapping Function Revision** Recall the definition of PLM (Definition C.1) and DLM (Definition C.2) are
> - PLM: $\mathrm{Acc}(f_{\rm plm}) = \sum_{y^{\rm T} \in \mathcal{Y}^{\rm T}} p(y^{\rm T}) \cdot \sum_{y^{\rm S} \in \mathcal{Y}^{\rm S}} p(y^{\rm S}) \cdot \omega_{y^{\rm S}, y^{\rm T}}$
> - DLM: $\mathrm{Acc}(f_{\rm dlm}) = \sum_{y^{\rm T} \in \mathcal{Y}^{\rm T}} p(y^{\rm T}) \cdot \sum_{y^{\rm S} \in \mathcal{Y}^{\rm S}} p(y^{\rm S}) \cdot \delta_{y^{\rm S}, g(y^{\rm S})}$
>
> For multi-class cases, we need to revise the mapping function from $f_{\rm lm}(y^{\rm S}) \in \lbrace y^{\rm T}, 1 - y^{\rm T} \rbrace$ in binary label spaces to align with $\mathcal{Y}^{\rm S} = \lbrace1, 2, ..., k_{\rm S} \rbrace$, $\mathcal{Y}^{\rm T} = \lbrace1, 2, ..., k_{\rm T} \rbrace$. Thus, we need to expand previously used identity/flip mapping rule $g(y^{\rm S})$ to cover a broader range of possible mappings.
>
> **Proof Sketch** PLM can achieve at least the accuracy of the optimal DLM by constructing $\omega_{y^{\rm S}, y^{\rm T}}$ to match the optimal deterministic mapping rule $g^*(y^{\rm S})$, which ensures $\sum_{y^{\rm S} \in \mathcal{Y}^{\rm S}} p(y^{\rm S}) \cdot \omega_{y^{\rm S}, y^{\rm T}} \geq \sum_{y^{\rm S} \in \mathcal{Y}^{\rm S}} p(y^{\rm S}) \cdot \mathbb{I}[g^*(y^{\rm S}) = y^{\rm T}], \forall y^{\rm T} \in \mathcal{Y}^{\rm T}$. Intuitively, this inequality holds, especially when $g^*(y^{\rm S}) \neq y^{\rm T}$, $\mathbb{I}[g^*(y^{\rm S}) = y^{\rm T}]=0$, while $\omega_{y^{\rm S}, y^{\rm T}}$ can be greater than 0 in such cases, showing better flexibility.
>
> Due to the character limit, we leave the complete proof to future work. We hope this clarification addresses your concern.
>
> **W3:** Thank you. In Appendix Sec. E, we have acknowledged that optimal values may vary across datasets (Fig. 8-9). Our initial use of universal hyperparameters was intended to show that BLM/BLM+'s performance gains are not sensitive to hyperparameters.
>
> Following your advice, we quickly run additional experiments using a 70\%/30\% train/validation split of the original training set to find optimal hyperparameters for each dataset, shown as
>
> | | Flowers102 | UCF101 | DTD | OxfordPets | CIFAR10 |
> |------------|------------|--------|-------|------------|---------|
> | Optimal $\alpha$ | 0.15 | 0.15 | 0.5 | 0.5 | 0.5 |
> | Optimal $\lambda$ | 1 | 1 | 1 | 10 | 10 |
> | Accuracy (Trained on 70% Samples) | 45.82 | 31.84 | **43.75** | **72.27** | **66.54** |
> | Shared $\alpha$ | 0.15 | 0.15 | 0.15 | 0.15 | 0.15 |
> | Shared $\lambda$ | 1 | 1 | 1 | 1 | 1 |
> | Accuracy (Trained on 70% Samples) | 45.82 | 31.84 | 42.31 | 70.52 | 66.04 |
>
> Observe that dataset-specific tuning indeed yields better performance compared to shared hyperparameters, suggesting that optimal hyperparameters tailored to each dataset are desired. We plan to include these in the revision.
>
> **W4:** Thanks for your question. This observation stems from the statistical nature of our BLM+ algorithm, which computes label mappings based on the co-occurrence of predicted pretrained labels and ground truth downstream labels throughout the VR learning iterations (Fig. 4 visualizes the top-3 predicted pretrained label at each iteration).
>
> In the later stages of VR learning, we observe that for marigold images $X^\mathrm{T}$, the conditional probabilities shifted:
> $p(Y^{\rm T}={\tt Marigold}|Y^{\rm S}={\tt Pineapple}, X^{\rm T})>p(Y^{\rm T}={\tt Marigold}|Y^{\rm S}={\tt Teddy}, X^{\rm T})$
>
> $p(Y^{\rm T}={\tt Marigold}|Y^{\rm S}={\tt Pineapple}, X^{\rm T})>p(Y^{\rm T}={\tt Marigold}|Y^{\rm S}={\tt Broccoli}, X^{\rm T})$
>
> This indicates that the feature of marigold images, after processing by input VR and the pretrained model, share more similarities with pineapple than with teddy bears or broccoli. Thus, pineapple replaced these other labels among the top-k predicted pretrained labels.
>
> Visually, this replacement is intuitive as well: The colors of a pineapple — primarily yellow, orange, and gold — are similar to those of Marigold. Teddy bear and Airedale are predominantly brown, while guacamole and broccoli are mainly green. Additionally, the shape of a pineapple, which is typically oval, resembles that of Marigold.
>
> **W5:** Thank you for your suggestion. While space limitations led us to initially place the algorithm tables in the appendix (lines 455 and 462), we recognize the importance of making this information more accessible. In the revision, we will prioritize incorporating them into the main text. We will also work on enriching and smoothing out the method presentation to make it more engaging.
>
> **W6:** Yes, BLM+ without Top-K and Bayes is the same as BLM without Bayes. Thanks for the reminder, we will explain it clearly.
>
> **W7:** Thank you for your questions. As we follow the settings in [1], the fine-tuning results can be directly quoted from [1], and we will include this in the next version of our paper. Regarding training time, please refer to **Common Question 1** for a detailed answer. Regarding the standard to determine when VR is needed, we believe that (1) when issues such as copyright or avoiding catastrophic forgetting exist, the pre-trained model needs to be kept unchanged; (2) when the resources for training downstream tasks are limited, VR can be used.
>
> [1] Chen et al. Understanding and improving visual prompting: A label-mapping perspective. In CVPR, 2023.

---

### Author Rebuttal · Authors · 2024-08-07

**Common Question 1:** Concerns about the required number of epochs and training time for BLM/BLM+

**Response 1:** Regarding the number of epochs, we initially used 200 epochs as with the original papers to ensure a fair comparison with the baseline methods. However, during the rebuttal stage, we conducted additional experiments to assess the impact of different epoch numbers (60, 100, 150) on our BLM/BLM+ model, using ResNet-18 as the pretrained model. The results are shown in Table 1.
**Table 1. Epoch Numbers and Testing Accuracy of Different Methods**
|    |    |BLM| (ours)|    |  |  BLM+    | (ours) |      || ILM  | FLM  |
|---------------------|------|------|------|------|--------------|------|------|------|---|------|------|
| Epochs              | 60          | 100  | 150  | 200  | 60           | 100  | 150  | 200  || 200  | 200  |
| Average on 12 Tasks (%) | 44.5        | 45.2 | 45.5 | 45.3 | 45.8         | 46.4 | 46.9 | 46.7 || 40.6 | 37.2 |

We found that running 100 epochs yields results comparable to those achieved with 200 epochs. This demonstrates that BLM and BLM+ require **less convergence time**, highlighting their efficiency.

Training time for one epoch is listed in Table 6 of our paper submission. Additionally, the total training time for one task, in comparison with baseline visual reprogramming methods and fine-tuning methods, is calculated below:

**Table 2. Time Consumption of Different Methods on Flowers102 Dataset (Single A100 GPU)**
|             |                  | VR: Baselines |       |    VR: Ours   |                |      VR: Ours             |                    |   Finetuning Methods  |                  |
|-------------|------------------|:-------------:|:-----:|:-----------------:|:------------------:|:-----------------:|:------------------:|:---------------------:|:----------------:|
|             |                  | FLM           | ILM   | BLM（200 epochs） | BLM+（200 epochs） | **BLM（100 epochs)** | **BLM+（100 epochs)** | Finetuning Last Layer | Fully Finetuning |
|  ResNet-18  | Parameter Number | 0.10M         | 0.10M | 0.10M             | 0.10M              | 0.10M             | 0.10M              | 0.51M                 | 11.7M            |
|             | Training Time (min) | 11.97         | 12.04 | 11.95             | 13.06              | **6.03**              | **6.52**               | 14.03                 | 15.28            |
| ResNeXt-101 | Parameter Number | 0.10M         | 0.10M | 0.10M             | 0.10M              | 0.10M             | 0.10M              | 2.0M                  | 88.8M            |
|             | Training Time (min) | 24.68         | 24.81 | 24.51             | 24.71              | **12.33**             | **12.44**              | 24.49                | 35.07            |

Combined with the results in the table, we analyze the efficiency of BLM and BLM+ from three perspectives:

(1) **Extra Consumption of Calculating Mapping Matrix $\omega$ Compared with One-to-One Mapping:** Compared to the baseline method ILM, the additional cost for BLM/BLM+ primarily involves the gradient-free multiplication and division within the mapping matrix (which is sized according to the source and target label spaces, 1000 × 102 in this case). This additional cost is minimal, as shown by the 4th-6th columns in Table 2.

(2) **Time Consumption of Updating Mapping Matrix $\omega$ per Epoch:** The baseline method FLM calculates the mapping $\omega$ once and keeps it fixed, while ILM and our methods update $\omega$ at each step. However, updating $\omega$ does not require running the model to obtain current predictions each epoch. Instead, predictions from the most recent epoch can be reused. As a result, there is no noticeable time overhead for updating $\omega$ per epoch, as indicated by the 3rd-6th columns in Table 2.

(3) **Time Consumption Compared with Finetuning Methods:** Since BLM/BLM+ use **fewer parameters** and **converge in fewer epochs**, they are significantly faster than finetuning the last layer or the entire model. This is demonstrated in the 7th-10th columns of Table 2.

---

**Common Question 2:** Detailed Explanations of Conditional Probabilities in Section 4.1

**Response 2:** Eq. (6) and (7) aim to estimate $p(Y^{\rm T} = y^{\rm T}, Y^{\rm S} = y^{\rm S} \mid X^{\rm T})$ and $p(Y^{\rm S} = y^{\rm S} \mid X^{\rm T})$ respectively. Here, $X^{\rm T} \in \mathcal{X}^{\rm T}$ represents a variable from the downstream task input space, while $Y^{\rm T} \in \mathcal{Y}^{\rm T}$ and $Y^{\rm S} \in \mathcal{Y}^{\rm S}$ are variables from the target and source label spaces, respectively.

The conditional probability $p(Y^{\rm T} = y^{\rm T}, Y^{\rm S} = y^{\rm S} \mid X^{\rm T})$ represents the joint distribution of $Y^{\rm T}$ and $Y^{\rm S}$, given the input reprogramming $f_{\rm in}$, the pretrained model $f_{\rm pre}$, and the variable $X^{\rm T}$ of the downstream task. Similarly, $p(Y^{\rm S} = y^{\rm S} \mid X^{\rm T})$ represents the distribution of $Y^{\rm S}$ under these conditions.

For example, consider the following setup:
- $\mathcal{Y}^{\rm T} = \lbrace\tt Cat, \tt Dog\rbrace$,
- $\mathcal{Y}^{\rm S} = \lbrace\tt CockerSpaniel, \tt EnglishSpringer, \tt EgyptianCat\rbrace$,
- Downstream samples are $ \lbrace(x_1, {\tt Dog}), (x_2, {\tt Dog}), (x_3, {\tt Dog}), (x_4, {\tt Cat})\rbrace$.

If the reprogrammed predictions calculated by $f_{\rm pre}(f_{\rm in}(x_i \mid \theta))$ are $\lbrace x_1: {\tt CockerSpaniel}, x_2: {\tt CockerSpaniel}, x_3: {\tt EnglishSpringer}, x_4: {\tt EgyptianCat}\rbrace$, then $p(Y^{\rm T} = y^{\rm T}, Y^{\rm S} = y^{\rm S} \mid X^{\rm T})$ can be estimated as a 2 \* 3 matrix with the following nonzero values:
- $p(Y^{\rm T} = {\tt Dog}, Y^{\rm S} = {\tt CockerSpaniel} \mid X^{\rm T}) = \frac{1}{2}$,
- $p(Y^{\rm T} = {\tt Dog}, Y^{\rm S} = {\tt EnglishSpringer} \mid X^{\rm T}) = \frac{1}{4}$,
- $p(Y^{\rm T} = {\tt Cat}, Y^{\rm S} = {\tt EgyptianCat} \mid X^{\rm T}) = \frac{1}{4}$.

---

### Decision · Program_Chairs · 2024-09-25

**Decision:**

Accept (oral)

**Comment:**

This paper presents work on visual reprogramming -- leveraging existing vision models and adapting them to solve tasks involving different labels.  The core contribution is a Bayesian approach to label mapping that alleviates shortcomings of previous (one-to-one mapping) approaches, and better utilizes cross-category information in conducting label mapping.

The reviewers noted the clear strengths of the paper in this regard, with a clear motivation, solid execution, and impressive empirical results.  A few minor issues were raised in the reviews, which were largely addressed in the author responses.

Visual reprogramming, and more broadly adapting pre-trained models to solve new tasks, is an active, important area of research.  This paper makes solid contributions to this literature, and as such is recommended for acceptance to NeurIPS.